# Learning Overparameterized Neural Networks via Stochastic Gradient Descent on Structured Data

**Yuanzhi Li**
Computer Science Department
Stanford University
Stanford, CA 94305
yuanzhil@stanford.edu

**Yingyu Liang**
Department of Computer Sciences
University of Wisconsin-Madison
Madison, WI 53706
yliang@cs.wisc.edu

## Abstract

Neural networks have many successful applications, while much less theoretical understanding has been gained. Towards bridging this gap, we study the problem of learning a two-layer overparameterized ReLU neural network for multi-class classification via stochastic gradient descent (SGD) from random initialization. In the overparameterized setting, when the data comes from mixtures of well-separated distributions, we prove that SGD learns a network with a small generalization error, albeit the network has enough capacity to fit arbitrary labels. Furthermore, the analysis provides interesting insights into several aspects of learning neural networks and can be verified based on empirical studies on synthetic data and on the MNIST dataset.

## 1 Introduction

Neural networks have achieved great success in many applications, but despite a recent increase of theoretical studies, much remains to be explained. For example, it is empirically observed that learning with stochastic gradient descent (SGD) in the overparameterized setting (i.e., learning a large network with number of parameters larger than the number of training data points) does not lead to overfitting [24, 31]. Some recent studies use the low complexity of the learned solution to explain the generalization, but usually do not explain how the SGD or its variants favors low complexity solutions (i.e., the inductive bias or implicit regularization) [3, 23]. It is also observed that overparameterization and proper random initialization can help the optimization [28, 12, 26, 18], but it is also not well understood why a particular initialization can improve learning. Moreover, most of the existing works trying to explain these phenomenons in general rely on unrealistic assumptions about the data distribution, such as Gaussian-ness and/or linear separability [32, 25, 10, 17, 7].

This paper thus proposes to study the problem of learning a two-layer overparameterized neural network using SGD for classification, on data with a more realistic structure. In particular, the data in each class is a mixture of several components, and components from different classes are well separated in distance (but the components in each class can be close to each other). This is motivated by practical data. For example, on the dataset MNIST [15], each class corresponds to a digit and can have several components corresponding to different writing styles of the digit, and an image in it is a small perturbation of one of the components. On the other hand, images that belong to the same component are closer to each other than to an image of another digit. Analysis in this setting can then help understand how the structure of the practical data affects the optimization and generalization.

In this setting, we prove that when the network is sufficiently overparameterized, SGD provably learns a network close to the random initialization and with a small generalization error. This result shows that in the overparameterized setting and when the data is well structured, though in principle

the network can overfit, SGD with random initialization introduces a strong inductive bias and leads to good generalization.

Our result also shows that the overparameterization requirement and the learning time depends on the parameters inherent to the structure of the data but not on the ambient dimension of the data. More importantly, the analysis to obtain the result also provides some interesting theoretical insights for various aspects of learning neural networks. It reveals that the success of learning crucially relies on overparameterization and random initialization. These two combined together lead to a tight coupling around the initialization between the SGD and another learning process that has a benign optimization landscape. This coupling, together with the structure of the data, allows SGD to find a solution that has a low generalization error, while still remains in the aforementioned neighborhood of the initialization. Our work makes a step towrads explaining how overparameterization and random initialization help optimization, and how the inductive bias and good generalization arise from the SGD dynamics on structured data. Some other more technical implications of our analysis will be discussed in later sections, such as the existence of a good solution close to the initialization, and the low-rankness of the weights learned. Complementary empirical studies on synthetic data and on the benchmark dataset MNIST provide positive support for the analysis and insights.

## 2   Related Work

**Generalization of neural networks.** Empirical studies show interesting phenomena about the generalization of neural networks: practical neural networks have the capacity to fit random labels of the training data, yet they still have good generalization when trained on practical data [24, 31, 2]. These networks are overparameterized in that they have more parameters than statistically necessary, and their good generalization cannot be explained by naïvely applying traditional theory. Several lines of work have proposed certain low complexity measures of the learned network and derived generalization bounds to better explain the phenomena. [3, 23, 21] proved spectrally-normalized margin-based generalization bounds, [9, 23] derived bounds from a PAC-Bayes approach, and [1, 33, 4] derived bounds from the compression point of view. They, in general, do not address why the low complexity arises. This paper takes a step towards this direction, though on two-layer networks and a simplified model of the data.

**Overparameterization and implicit regularization.** The training objectives of overparameterized networks in principle have many (approximate) global optima and some generalize better than the others [14, 8, 2], while empirical observations imply that the optimization process in practice prefers those with better generalization. It is then an interesting question how this implicit regularization or inductive bias arises from the optimization and the structure of the data. Recent studies are on SGD for different tasks, such as logistic regression [27] and matrix factorization [11, 19, 16]. More related to our work is [7], which studies the problem of learning a two-layer overparameterized network on linearly separable data and shows that SGD converges to a global optimum with good generalization. Our work studies the problem on data with a well clustered (and potentially not linearly separable) structure that we believe is closer to practical scenarios and thus can advance this line of research.

**Theoretical analysis of learning neural networks.** There also exists a large body of work that analyzes the optimization landscape of learning neural networks [13, 26, 30, 10, 25, 29, 6, 32, 17, 5]. They in general need to assume unrealistic assumptions about the data such as Gaussian-ness, and/or have strong assumptions about the network such as using only linear activation. They also do not study the implicit regularization by the optimization algorithms.

## 3   Problem Setup

In this work, a two-layer neural network with ReLU activation for $k$-classes classification is given by $f = (f_1, f_2, \cdots, f_k)$ such that for each $i \in [k]$:

$$f_i(x) = \sum_{r=1}^{m} a_{i,r} \mathbf{ReLU}(\langle w_r, x \rangle)$$

where $\{w_r \in \mathbb{R}^d\}$ are the weights for the $m$ neurons in the hidden layer, $\{a_{i,r} \in \mathbb{R}\}$ are the weights of the top layer, and $\mathbf{ReLU}(z) = \max\{0, z\}$.

**Assumptions about the data.** The data is generated from a distribution $\mathcal{D}$ as follows. There are $k \times l$ unknown distributions $\{\mathcal{D}_{i,j}\}_{i \in [k], j \in [l]}$ over $\mathbb{R}^d$ and probabilities $p_{i,j} \geq 0$ such that $\sum_{i,j} p_{i,j} = 1$. Each data point $(x, y)$ is i.i.d. generated by: (1) Sample $z \in [k] \times [l]$ such that $\Pr[z = (i, j)] = p_{i,j}$; (2) Set label $y = z[0]$, and sample $x$ from $\mathcal{D}_z$. Assume we sample $N$ points $\{(x_i, y_i)\}_{i=1}^N$.

Let us define the support of a distribution $\mathcal{D}$ with density $p$ over $\mathcal{R}^d$ as $\text{supp}(\mathcal{D}) = \{x : p(x) > 0\}$, the distance between two sets $\mathcal{S}_1, \mathcal{S}_2 \subseteq \mathcal{R}^d$ as $\text{dist}(\mathcal{S}_1, \mathcal{S}_2) = \min_{x \in \mathcal{S}_1, y \in \mathcal{S}_2}\{\|x - y\|_2\}$, and the diameter of a set $\mathcal{S}_1 \subseteq \mathcal{R}^d$ as $\text{diam}(\mathcal{S}_1) = \max_{x,y \in \mathcal{S}_1}\{\|x - y\|_2\}$. Then we are ready to make the assumptions about the data.

**(A1)** (Separability) There exists $\delta > 0$ such that for every $i_1 \neq i_2 \in [k]$ and every $j_1, j_2 \in [l]$, $\text{dist}\left(\text{supp}(\mathcal{D}_{i_1,j_1}), \text{supp}(\mathcal{D}_{i_2,j_2})\right) \geq \delta$. Moreover, for every $i \in [k], j \in [l]$,[1] $\text{diam}(\text{supp}(\mathcal{D}_{i,j})) \leq \lambda\delta$, for $\lambda \leq 1/(8l)$.

**(A2)** (Normalization) Any $x$ from the distribution has $\|x\|_2 = 1$.

A few remarks are worthy. Instead of having one distribution for one class, we allow an arbitrary $l \geq 1$ distributions in each class, which we believe is a better fit to the real data. For example, in MNIST, a class can be the number 1, and $l$ can be the different styles of writing 1 (1 or | or /).

Assumption **(A2)** is for simplicity, while **(A1)** is our key assumption. With $l \geq 1$ distributions inside each class, our assumption allows data that is not linearly separable, e.g., XOR type data in $\mathcal{R}^2$ where there are two classes, one consisting of two balls of diameter $1/10$ with centers $(0, 0)$ and $(2, 2)$ and the other consisting of two of the same diameter with centers $(0, 2)$ and $(2, 0)$. See Figure 3 in Appendix C for an illustration. Moreover, essentially the only assumption we have here is $\lambda = O(1/l)$. When $l = 1$, $\lambda = O(1)$, which is the minimal requirement on the order of $\lambda$ for the distribution to be efficiently learnable. Our work allows larger $l$, so that the data can be more complicated inside each class. In this case, we require the separation to also be higher. When we increase $l$ to refine the distributions inside each class, we should expect the diameters of each distribution become smaller as well. As long as the rate of diameter decreasing in each distribution is greater than the total number of distributions, then our assumption will hold.

**Assumptions about the learning process.** We will only learn the weight $w_r$ to simplify the analysis. Since the ReLU activation is positive homogeneous, the effect of overparameterization can still be studied, and a similar approach has been adopted in previous work [7]. So the network is also written as $y = f(x, w) = (f_1(x, w), \cdots, f_k(x, w))$ for $w = (w_1, \cdots, w_r)$.

We assume the learning is from a random initialization:

**(A3)** (Random initialization) $w_r^{(0)} \sim \mathcal{N}(0, \sigma^2 \mathbf{I})$, $a_{i,r} \sim \mathcal{N}(0, 1)$, with $\sigma = \frac{1}{m^{1/2}}$.

The learning process minimizes the cross entropy loss over the softmax, defined as:

$$L(w) = -\frac{1}{N} \sum_{s=1}^N \log o_{y_s}(x_s, w), \text{ where } o_y(x, w) = \frac{e^{f_y(x,w)}}{\sum_{i=1}^k e^{f_i(x,w)}}.$$

Let $L(w, x_s, y_s) = -\log o_{y_s}(x_s, w)$ denote the cross entropy loss for a particular point $(x_s, y_s)$.

We consider a minibatch SGD of batch size $B$, number of iterations $T = N/B$ and learning rate $\eta$ as the following process: Randomly divide the total training examples into $T$ batches, each of size $B$. Let the indices of the examples in the $t$-th batch be $\mathcal{B}_t$. At each iteration, the update is[2]

$$w_r^{(t+1)} = w_r^{(t)} - \eta \frac{1}{B} \sum_{s \in \mathcal{B}_t} \frac{\partial L(w^{(t)}, x_s, y_s)}{\partial w_r^{(t)}}, \forall r \in [m], \text{ where}$$

$$\frac{\partial L(w, x_s, y_s)}{\partial w_r} = \left( \sum_{i \neq y_s} a_{i,r} o_i(x_s, w) - \sum_{i \neq y_s} a_{y_s,r} o_i(x_s, w) \right) 1_{\langle w_r, x_s \rangle \geq 0} x_s. \quad (1)$$

# 4 Main Result

For notation simplicity, for a target error $\varepsilon$ (to be specified later), with high probability (or w.h.p.) means with probability $1 - 1/\text{poly}(1/\delta, k, l, m, 1/\varepsilon)$ for a sufficiently large polynomial poly, and $\tilde{O}$ hides factors of $\text{poly}(\log 1/\delta, \log k, \log l, \log m, \log 1/\varepsilon)$.

**Theorem 4.1.** *Suppose the assumptions (A1)(A2)(A3) are satisfied. Then for every $\varepsilon > 0$, there is $M = poly(k, l, 1/\delta, 1/\varepsilon)$ such that for every $m \geq M$, after doing a minibatch SGD with batch size $B = poly(k, l, 1/\delta, 1/\varepsilon, \log m)$ and learning rate $\eta = \frac{1}{m \cdot poly(k, l, 1/\delta, 1/\varepsilon, \log m)}$ for $T = poly(k, l, 1/\delta, 1/\varepsilon, \log m)$ iterations, with high probability:*

$$\Pr_{(x,y) \sim \mathcal{D}} \left[ \forall j \in [k], j \neq y, f_y(x, w^{(T)}) > f_j(x, w^{(T)}) \right] \geq 1 - \varepsilon.$$

Our theorem implies if the data satisfies our assumptions, and we parametrize the network properly, then we only need polynomial in $k, l, 1/\delta$ many samples to achieve a good prediction error. This error is measured directly on the true distribution $\mathcal{D}$, not merely on the input data used to train this network. Our result is also dimension free: There is no dependency on the underlying dimension $d$ of the data, the complexity is fully captured by $k, l, 1/\delta$. Moreover, no matter how much the network is overparameterized, it will only increase the total iterations by factors of $\log m$. So we can overparameterize by an *sub-exponential amount* without significantly increasing the complexity.

Furthermore, we can always treat each input example as an individual distribution, thus $\lambda$ is always zero. In this case, if we use batch size $B$ for $T$ iterations, we would have $l = N = BT$. Then our theorem indicate that as long as $m = \text{poly}(N, 1/\delta')$, where $\delta'$ is the minimal distance between each examples, we can actually fit arbitrary labels of the input data. However, since the total iteration only depends on $\log m$, when $m = \text{poly}(N, 1/\delta')$ but the input data is actually structured (with small $k, l$ and large $\delta$), then SGD can actually achieve a small generalization error, *even when* the network has enough capacity to fit arbitrary labels of the training examples (and can also be done by SGD). Thus, we prove that SGD has a strong inductive bias on structured data: Instead of finding a bad global optima that can fit arbitrary labels, it actually finds those with good generalization guarantees. This gives more thorough explanation to the empirical observations in [24, 31].

# 5 Intuition and Proof Sketch for A Simplified Case

To train a neural network with ReLU activations, there are two questions need to be addressed:

1. Why can SGD optimize the training loss? Or even finding a critical point? Since the underlying network is highly non-smooth, existing theorems do not give any finite convergence rate of SGD for training neural network with ReLUs activations.

2. Why can the trained network generalize? Even when the capacity is large enough to fit random labels of the input data? This is known as the inductive bias of SGD.

This work takes a step towards answering these two questions. We show that when the network is overparameterized, it becomes more "pseudo smooth", which makes it easir for SGD to minimize the training loss, and furthermore, it will not hurt the generalization error. Our proof is based on the following important observation:

> The more we overparameterize the network, the less likely the activation pattern for one neuron and one data point will change in a fixed number of iterations.

This observation allows us to couple the gradient of the true neural network with a "pseudo gradient" where the activation pattern for each data point and each neuron is fixed. That is, when computing the "pseudo gradient", for fixed $r, i$, whether the $r$-th hidden node is activated on the $i$-th data point $x_i$ will always be the same for different $t$. (But for fixed $t$, for different $r$ or $i$, the sign can be different.) We are able to prove that unless the generalization error is small, the "pseudo gradient" will always be large. Moreover, we show that the network is actually smooth thus SGD can minimize the loss.

We then show that when the number $m$ of hidden neurons increases, with a properly decreasing learning rate, the total number of iterations it takes to minimize the loss is roughly not changed. However, the total number of iterations that we can couple the true gradient with the pseudo one

increases. Thus, there is a polynomially large $m$ so that we can couple these two gradients until the network reaches a small generalization error.

## 5.1 A Simplified Case: No Variance

Here we illustrate the proof sketch for a simplified case and Appendix A provides the proof. The proof for the general case is provided in Appendix B. In the simplified case, we further assume:

**(S)** (No variance) Each $\mathcal{D}_{a,b}$ is a single data point $(x_{a,b}, a)$, and also we are doing full batch gradient descent as opposite to the minibatch SGD.

Then we reload the loss notation as $L(w) = \sum_{a \in [k], b \in [l]} p_{a,b} L(w, x_{a,b}, a)$, and the gradient is

$$\frac{\partial L(w)}{\partial w_r} = \sum_{a \in [k], b \in [l]} p_{a,b} \left( \sum_{i \neq a} a_{i,r} o_i(x_{a,b}, w) - \sum_{i \neq a} a_{a,r} o_i(x_{a,b}, w) \right) 1_{\langle w_r, x_{a,b} \rangle \geq 0} x_{a,b}.$$

Following the intuition above, we define the pseudo gradient as

$$\frac{\tilde{\partial} L(w)}{\partial w_r} = \sum_{a \in [k], b \in [l]} p_{a,b} \left( \sum_{i \neq a} a_{i,r} o_i(x_{a,b}, w) - \sum_{i \neq a} a_{a,r} o_i(x_{a,b}, w) \right) 1_{\langle w_r^{(0)}, x_{a,b} \rangle \geq 0} x_{a,b},$$

where it uses $1_{\langle w_r^{(0)}, x_{a,b} \rangle \geq 0}$ instead of $1_{\langle w_r, x_{a,b} \rangle \geq 0}$ as in the true gradient. That is, the activation pattern is set to be that in the initialization. Intuitively, the pseudo gradient is similar to the gradient for a pseudo network $g$ (but not exactly the same), defined as $g_i(x, w) := \sum_{r=1}^{m} a_{i,r} \langle w_r, x \rangle 1_{\langle w_r^{(0)}, x \rangle \geq 0}$.

Coupling the gradients is then similar to coupling the networks $f$ and $g$.

For simplicity, let $v_{a,a,b} := \sum_{i \neq a} o_i(x_{a,b}, w) = \frac{\sum_{i \neq a} e^{f_i(x_{a,b}, w)}}{\sum_{i=1}^{k} e^{f_i(x_{a,b}, w)}}$ and when $s \neq a$, $v_{s,a,b} := -o_s(x_{a,b}, w) = -\frac{e^{f_s(x_{a,b}, w)}}{\sum_{i=1}^{k} e^{f_i(x_{a,b}, w)}}$. Roughly, if $v_{a,a,b}$ is small, then $f_a(x_{a,b}, w)$ is relatively larger compared to the other $f_i(x_{a,b}, w)$, so the classification error is small.

We prove the following two main lemmas. The first says that at each iteration, the total number of hidden units whose gradient can be coupled with the pseudo one is quite large.

**Lemma 5.1** (Coupling). *W.h.p. over the random initialization, for every $\tau > 0$, for every $t = \tilde{O}\left(\frac{\tau}{\eta}\right)$, we have that for at least $1 - \frac{e\tau kl}{\sigma}$ fraction of $r \in [m]$: $\frac{\partial L(w^{(t)})}{\partial w_r} = \frac{\tilde{\partial} L(w^{(t)})}{\partial w_r}$.*

The second lemma says that the pseudo gradient is large unless the error is small.

**Lemma 5.2.** *For $m = \tilde{\Omega}\left(\frac{k^3 l^2}{\delta}\right)$, for every $\{p_{a,b} v_{i,a,b}\}_{i,a \in [k], b \in [l]} \in [-v, v]$ (that depends on $w_r^{(0)}, a_{i,r},$ etc.) with $\max\{p_{a,b} v_{i,a,b}\}_{i,a \in [k], b \in [l]} = v$, there exists at least $\Omega(\frac{\delta}{kl})$ fraction of $r \in [m]$ such that $\left\| \frac{\tilde{\partial} L(w)}{\partial w_r} \right\|_2 = \tilde{\Omega}\left(\frac{v\delta}{kl}\right)$.*

We now illustrate how to use these two lemmas to show the convergence for a small enough learning rate $\eta$. For simplicity, let us assume that $kl/\delta = O(1)$ and $\varepsilon = o(1)$. Thus, by Lemma 5.2 we know that unless $v \leq \varepsilon$, there are $\Omega(1)$ fraction of $r$ such that $\left\| \tilde{\partial} L(w)/\partial w_r \right\|_2 = \Omega(\varepsilon)$. Moreover, by Lemma 5.1 we know that we can pick $\tau = \Theta(\sigma\varepsilon)$ so $e\tau/\sigma = \Theta(\varepsilon)$, which implies that there are $\Omega(1)$ fraction of $r$ such that $\|\partial L(w)/\partial w_r\|_2 = \Omega(\varepsilon)$ as well. For small enough learning rate $\eta$, doing one step of gradient descent will thus decrease $L(w)$ by $\Omega(\eta m \varepsilon^2)$, so it converges in $t = O\left(1/\eta m \varepsilon^2\right)$ iterations. In the end, we just need to make sure that $1/\eta m \varepsilon^2 \leq O(\tau/\eta) = \Theta(\sigma\varepsilon/\eta)$ so we can always apply the coupling Lemma 5.1. By $\sigma = \tilde{O}(1/m^{-1/2})$ we know that this is true as long as $m \geq \text{poly}(1/\varepsilon)$. A small $v$ can be shown to lead to a small generalization error.

## 6 Discussion of Insights from the Analysis

Our analysis, though for learning two-layer networks on well structured data, also sheds some light upon learning neural networks in more general settings.

**Generalization.** Several lines of recent work explain the generalization phenomenon of overparameterized networks by low complexity of the learned networks, from the point views of spectrally-normalized margins [3, 23, 21], compression [1, 33, 4], and PAC-Bayes [9, 23].

Our analysis has partially explained how SGD (with proper random initialization) on structured data leads to the low complexity from the compression and PCA-Bayes point views. We have shown that in a neighborhood of the random initialization, w.h.p. the gradients are similar to those of another benign learning process, and thus SGD can reduce the error and reach a good solution while still in the neighborhood. The closeness to the initialization then means the weights (or more precisely the difference between the learned weights and the initialization) can be easily compressed. In fact, empirical observations have been made and connected to generalization in [22, 1]. Furthermore, [1] explicitly point out such a compression using a helper string (corresponding to the initialization in our setting). [1] also point out that the compression view can be regarded as a more explicit form of the PAC-Bayes view, and thus our intuition also applies to the latter.

The existence of a solution of a small generalization error near the initialization is itself not obvious. Intuitively, on structured data, the updates are structured signals spread out across the weights of the hidden neurons. Then for prediction, the random initialized part in the weights has strong cancellation, while the structured signal part in the weights collectively affects the output. Therefore, the latter can be much smaller than the former while the network can still give accurate predictions. In other words, there can be a solution not far from the initialization with high probability.

Some insight is provided on the low rank of the weights. More precisely, when the data are well clustered around a few patterns, the accumulated updates (difference between the learned weights and the initialization) should be approximately low rank, which can be seen from checking the SGD updates. However, when the difference is small compared to the initialization, the spectrum of the final weight matrix is dominated by that of the initialization and thus will tend to closer to that of a random matrix. Again, such observations/intuitions have been made in the literature and connected to compression and generalization (e.g., [1]).

**Implicit regularization v.s. structure of the data.** Existing work has analyzed the implicit regularization of SGD on logistic regression [27], matrix factorization [11, 19, 16], and learning two-layer networks on linearly separable data [7]. Our setting and also the analysis techniques are novel compared to the existing work. One motivation to study on structured data is to understand the role of structured data play in the implicit regularization, i.e., the observation that the solution learned on less structured or even random data is further away from the initialization. Indeed, our analysis shows that when the network size is fixed (and sufficiently overparameterized), learning over poorly structured data (larger $k$ and $\ell$) needs more iterations and thus the solution can deviate more from the initialization and has higher complexity. An extreme and especially interesting case is when the network is overparameterized so that in principle it can fit the training data by viewing each point as a component while actually they come from structured distributions with small number of components. In this case, we can show that it still learns a network with a small generalization error; see the more technical discussion in Section 4.

We also note that our analysis is under the assumption that the network is sufficiently overparameterized, i.e., $m$ is a sufficiently large polynomial of $k$, $\ell$ and other related parameters measuring the structure of the data. There could be the case that $m$ is smaller than this polynomial but is more than sufficient to fit the data, i.e., the network is still overparameterized. Though in this case the analysis still provides useful insight, it does not fully apply; see our experiments with relatively small $m$. On the other hand, the empirical observations [24, 31] suggest that practical networks are highly overparameterized, so our intuition may still be helpful there.

**Effect of random initialization.** Our analysis also shows how proper random initializations helps the optimization and consequently generalization. Essentially, this guarantees that w.h.p. for weights close to the initialization, many hidden ReLU units will have the same activation patterns (i.e., activated or not) as for the initializations, which means the gradients in the neighborhood look like those when the hidden units have fixed activation patterns. This allows SGD makes progress when the loss is large, and eventually learns a good solution. We also note that it is essential to carefully set the scale of the initialization, which is a extensively studied topic [20, 28]. Our initialization has a scale related to the number of hidden units, which is particularly useful when the network size is varying, and thus can be of interest in such practical settings.

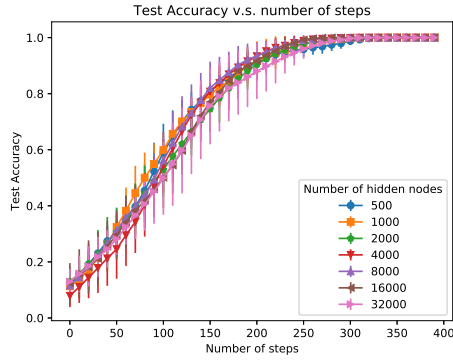

(a) Test accuracy

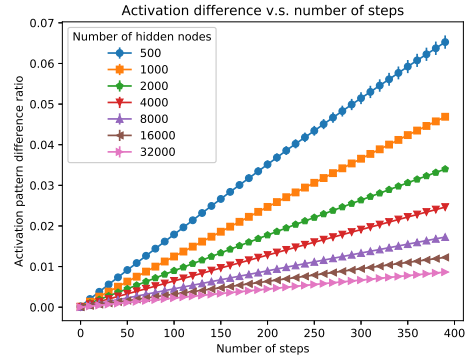

(b) Coupling

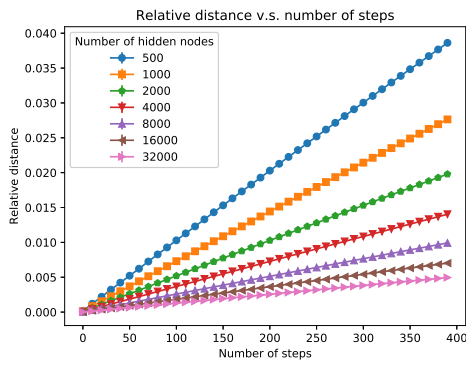

(c) Distance from the initialization

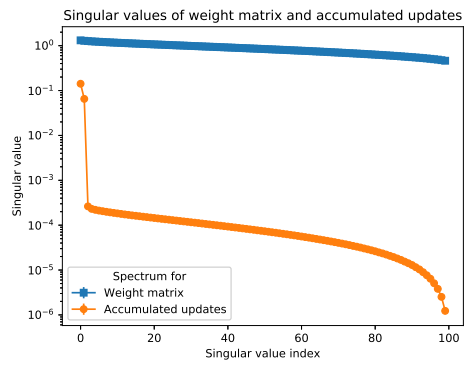

(d) Rank of accumulated updates ($y$-axis in log-scale)

Figure 1: Results on the synthetic data.

## 7 Experiments

This section aims at verifying some key implications: (1) the activation patterns of the hidden units couple with those at initialization; (2) The distance from the learned solution from the initialization is relatively small compared to the size of initialization; (3) The accumulated updates (i.e., the difference between the learned weight matrix and the initialization) have approximately low rank. These are indeed supported by the results on the synthetic and the MNIST data. Additional experiments are presented in Appendix D.

**Setup.** The synthetic data are of 1000 dimension and consist of $k = 10$ classes, each having $\ell = 2$ components. Each component is of equal probability $1/(kl)$, and is a Gaussian with covariance $\sigma^2/dI$ and its mean is i.i.d. sampled from a Gaussian distribution $\mathcal{N}(0, \sigma_0^2/d)$, where $\sigma = 1$ and $\sigma_0 = 5$. 1000 training data points and 1000 test data points are sampled.

The network structure and the learning process follow those in Section 3; the number of hidden units $m$ varies in the experiments, and the weights are initialized with $\mathcal{N}(0, 1/\sqrt{m})$. On the synthetic data, the SGD is run for $T = 400$ steps with batch size $B = 16$ and learning rate $\eta = 10/m$. On MNIST, the SGD is run for $T = 2 \times 10^4$ steps with batch size $B = 64$ and learning rate $\eta = 4 \times 10^3/m$.

Besides the test accuracy, we report three quantities corresponding to the three observations/implications to be verified. First, for coupling, we compute the fraction of hidden units whose activation pattern changed compared to the time at initialization. Here, the activation pattern is defined as 1 if the input to the ReLU is positive and 0 otherwise. Second, for distance, we compute the relative ratio $\|w^{(t)} - w^{(0)}\|_F / \|w^{(0)}\|_F$, where $w^{(t)}$ is the weight matrix at time $t$. Finally, for the rank of the accumulated updates, we plot the singular values of $w^{(T)} - w^{(0)}$ where $T$ is the final step. All experiments are repeated 5 times, and the mean and standard deviation are reported.

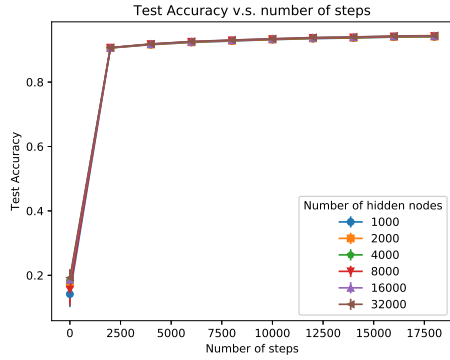

(a) Test accuracy

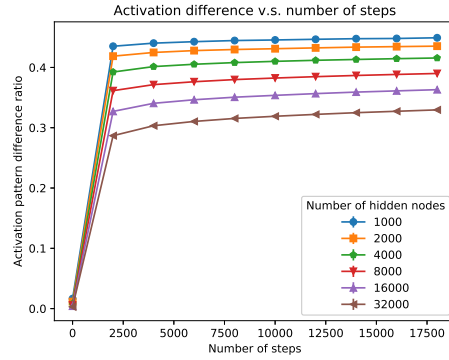

(b) Coupling

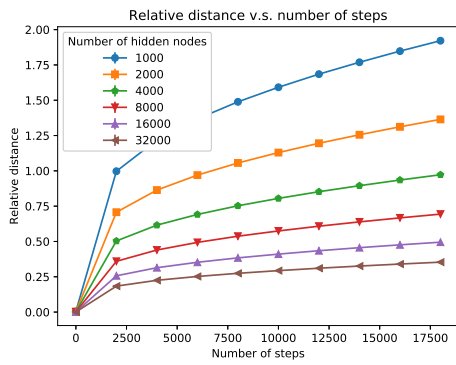

(c) Distance from the initialization

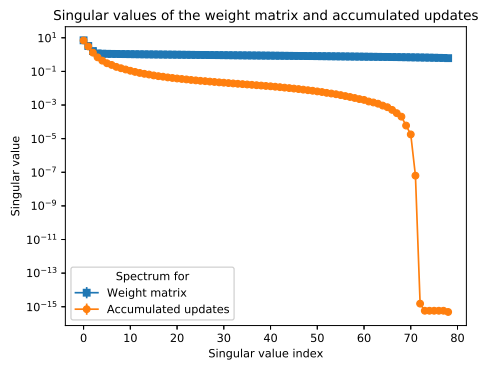

(d) Rank of accumulated updates ($y$-axis in log-scale)

Figure 2: Results on the MNIST data.

**Results.** Figure 1 shows the results on the synthetic data. The test accuracy quickly converges to $100\%$, which is even more significant with larger number of hidden units, showing that the overparameterization helps the optimization and generalization. Recall that our analysis shows that for a learning rate linearly decreasing with the number of hidden nodes $m$, the number of iterations to get the accuracy to achieve a desired accuracy should be roughly the same, which is also verified here. The activation pattern difference ratio is less than $0.1$, indicating a strong coupling. The relative distance is less than $0.1$, so the final solution is indeed close to the initialization. Finally, the top 20 singular values of the accumulated updates are much larger than the rest while the spectrum of the weight matrix do not have such structure, which is also consistent with our analysis.

Figure 2 shows the results on MNIST. The observation in general is similar to those on the synthetic data (though less significant), and also the observed trend become more evident with more overparameterization. Some additional results (e.g., varying the variance of the synthetic data) are provided in the appendix that also support our theory.

# 8 Conclusion

This work studied the problem of learning a two-layer overparameterized ReLU neural network via stochastic gradient descent (SGD) from random initialization, on data with structure inspired by practical datasets. While our work makes a step towards theoretical understanding of SGD for training neural networs, it is far from being conclusive. In particular, the real data could be separable with respect to different metric than $\ell_2$, or even a non-convex distance given by some manifold. We view this an important open direction.

## Acknowledgements

We would like to thank the anonymous reviewers of NIPS'18 and Jason Lee for helpful comments. This work was supported in part by FA9550-18-1-0166, NSF grants CCF-1527371, DMS-1317308, Simons Investigator Award, Simons Collaboration Grant, and ONR-N00014-16-1-2329. Yingyu Liang would also like to acknowledge that support for this research was provided by the Office of the Vice Chancellor for Research and Graduate Education at the University of Wisconsin Madison with funding from the Wisconsin Alumni Research Foundation.

## Footnotes

[1] The assumption $1/(8l)$ can be made to $1/[(1 + \alpha)l]$ for any $\alpha > 0$ by paying a large polynomial in $1/\alpha$ in the sample complexity. We will not prove it in this paper because we would like to highlight the key factors.

[2] Strictly speaking, $L(w, x_s, y_s)$ does not have gradient everywhere due to the non-smoothness of ReLU. One can view $\frac{\partial L(w, x_s, y_s)}{\partial w_r}$ as a convenient notation for the right hand side of (1).

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
