[Supplementary Material · supplementary.pdf]

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

# A Proofs for the Simplified Case

In the simplified case, we make the following simplifying assumption:

**(S)** (No variance) Each $\mathcal{D}_{a,b}$ is a single data point $(x_{a,b}, a)$, and also we are doing full batch gradient descent as opposite to the minibatch SGD.

Recall that the loss is then $L(w) = \sum_{a\in[k],b\in[l]} p_{a,b} L(w, x_{a,b}, a)$. The gradient descent update on $w$ is given by

$$w_r^{(t+1)} = w_r^{(t)} - \eta \frac{\partial L(w^{(t)})}{\partial w_r^{(t)}},$$

and the gradient is

$$\frac{\partial L(w)}{\partial w_r} = \sum_{a\in[k],b\in[l]} p_{a,b} \left( \sum_{i\neq a} a_{i,r} o_i(x_{a,b}, w) - \sum_{i\neq a} a_{a,r} o_i(x_{a,b}, w) \right) 1_{\langle w_r, x_{a,b}\rangle \geq 0} x_{a,b},$$

where $o_y(x, w) = \frac{e^{f_y(x,w)}}{\sum_{i=1}^k e^{f_i(x,w)}}$. The pseudo gradient is defined as

$$\frac{\tilde{\partial} L(w)}{\partial w_r} = \sum_{a\in[k],b\in[l]} p_{a,b} \left( \sum_{i\neq a} a_{i,r} o_i(x_{a,b}, w) - \sum_{i\neq a} a_{a,r} o_i(x_{a,b}, w) \right) 1_{\langle w_r^{(0)}, x_{a,b}\rangle \geq 0} x_{a,b}.$$

Let us call

$$v_{s,a,b}(w) = \begin{cases} \frac{\sum_{i\neq a} e^{f_i(x_{a,b},w)}}{\sum_{i=1}^k e^{f_i(x_{a,b},w)}} & \text{if } s = a; \\ -\frac{e^{f_s(x_{a,b},w)}}{\sum_{i=1}^k e^{f_i(x_{a,b},w)}} & \text{otherwise.} \end{cases}$$

When clear from the context, we write $v_{s,a,b}(w)$ as $v_{s,a,b}$. Then we can simplify the above expression as:

$$\frac{\tilde{\partial} L(w)}{\partial w_r} = \sum_{a\in[k],b\in[l],i\in[k]} p_{a,b} a_{i,r} v_{i,a,b} 1_{\langle w_r^{(0)}, x_{a,b}\rangle \geq 0} x_{a,b}.$$

By definition, $v_{i,a,b}$'s satisfy:

1. $\forall a \in [k], b \in [l] : v_{a,a,b} \in [0,1]$.

2. $\sum_{i=1}^k v_{i,a,b} = 0$.

Furthermore, $v_{a,a,b}$ indicates the "classification error". The smaller $v_{a,a,b}$ is, the smaller the classification error is.

In the following subsections, we first show that the gradient is coupled with the pseudo gradient, then show that if the classification error is large then the pseudo gradient is large, and finally prove the convergence.

## A.1 Coupling

We will show that $\partial L(w^{(t)})/\partial w_r$ is close to $\tilde{\partial} L(w^{(t)})/\partial w_r$ in the following sense:

**Lemma A.1** (Coupling, Lemma 5.1 restated). *W.h.p. over the random initialization, for every $\tau > 0$, for every $t = \tilde{O}\left(\frac{\tau}{\eta}\right)$, we have that for at least $1 - \frac{e\tau kl}{\sigma}$ fraction of $r \in [m]$:*

$$\frac{\partial L(w^{(t)})}{\partial w_r} = \frac{\tilde{\partial} L(w^{(t)})}{\partial w_r}.$$

*Proof.* W.h.p. we know that every $|a_{i,r}| \leq L = \tilde{O}(1)$. Thus, for every $r \in [m]$ and every $t \geq 0$, we have

$$\left\| \frac{\partial L(w^{(t)})}{\partial w_r} \right\|_2 \leq L$$

which implies that $\left\| w_r^{(t)} - w_r^{(0)} \right\|_2 \leq L\eta t$.

Now, for every $\tau \geq 0$, we consider the set $\mathcal{H}$ such that

$$\mathcal{H} = \left\{ r \in [m] \mid \forall a \in [k], b \in [l] : \left| \langle w_r^{(0)}, x_{a,b} \rangle \right| \geq \tau \right\}.$$

For every $r \in \mathcal{H}$ and every $t \leq \frac{\tau}{2L\eta}$, we know that for every $a \in [k], b \in [l]$:

$$\left| \langle w_r^{(t)} - w_r^{(0)}, x_{a,b} \rangle \right| \leq L\eta t \leq \frac{\tau}{2}$$

which implies that

$$\mathbb{1}_{\langle w_r^{(0)}, x_{a,b} \rangle \geq 0} = \mathbb{1}_{\langle w_r^{(t)}, x_{a,b} \rangle \geq 0}.$$

This implies that $\frac{\partial L(w^{(t)})}{\partial w_r} = \frac{\tilde{\partial} L(w^{(t)})}{\partial w_r}$.

Now, we need to bound the size of $\mathcal{H}$. Since $\langle w_r^{(0)}, x_{a,b} \rangle \sim \mathcal{N}(0, \sigma^2)$, by standard property of Gaussian we directly have that for $|\mathcal{H}| \geq 1 - \frac{e\tau kl}{\sigma}$. $\qquad \square$

## A.2 Error Large $\implies$ Gradient Large

The pseudo gradient can be rewritten as the following summation:

$$\frac{\tilde{\partial} L(w)}{\partial w_r} = \sum_{i \in [k]} a_{i,r} P_{i,r}$$

where

$$P_{i,r} = \sum_{a \in [k], b \in [l]} p_{a,b} v_{i,a,b} \mathbb{1}_{\langle w_r^{(0)}, x_{a,b} \rangle \geq 0} x_{a,b}.$$

We would like to show that if some $p_{a,b} v_{i,a,b}$ is large, a good fraction of $r \in [m]$ will have large pseudo gradient. Now, the first step is to show that for any fixed $\{p_{a,b} v_{i,a,b}\}$ (that does not depend on the random initialization $w_r^{(0)}$), with good probability (over the random choice of $w_r^{(0)}$) we have that $P_{i,r}$ is large; see Lemma A.2. Then we will take a union bound over an epsilon net on $\{p_{a,b} v_{i,a,b}\}$ to show that for every $\{p_{a,b} v_{ia,b}\}$ (that can depend on $w_r^{(0)}$), at least a good fraction of of $P_{i,r}$ is large; See Lemma A.3.

**Lemma A.2** (The geometry of **ReLU**). *For any possible fixed $\{p_{a,b} v_{1,a,b}\}_{a \in [k], b \in [l]} \in [-v, v]$ such that $p_{1,1} v_{1,1,1} = v$, we have:*

$$\Pr \left[ \|P_{1,r}\|_2 = \tilde{\Omega} \left( \frac{v\delta}{kl} \right) \right] = \Omega \left( \frac{\delta}{kl} \right).$$

Clearly, without **ReLU**, $P_{1,r}$ can be arbitrarily small if, say, $\forall b \in [l], v_{1,1,b} = v, p_{1,b} = p$ and $\sum_{b \in [l]} x_{1,b} = 0$. However, **ReLU** would prevent the cancellation of those $x_{1,b}$'s.

*Proof of Lemma A.2.* We will first prove that

$$h\left(w_r^{(0)}\right) = \sum_{a \in [k], b \in [l]} p_{a,b} v_{1,a,b} \textbf{ReLU} \left( \langle w_r^{(0)}, x_{a,b} \rangle \right) = \langle P_{1,r}, w_r^{(0)} \rangle$$

is large with good probability.

Let us decompose $w_r^{(0)}$ into:

$$w_r^{(0)} = \alpha x_{1,1} + \beta$$

where $\beta \perp x_{1,1}$. For every $\tau \geq 0$, consider the event $\mathcal{E}_\tau$ defined as

1. $|\alpha| \leq \tau$, and

2. for all $a \in [k]\backslash[1], b \in [l]$: $|\langle \beta, x_{a,b} \rangle| \geq 4\tau$.

By the definition of initialization $w_r^{(0)}$, we know that:
$$\alpha \sim \mathcal{N}(0, \sigma^2)$$

and

$$\langle \beta, x_{a,b} \rangle \sim \mathcal{N}(0, (1 - \langle x_{a,b}, x_{1,1} \rangle^2)\sigma^2)$$

By assumption we know that for every $a \in [k]\backslash[1], b \in [l]$:
$$1 - \langle x_{a,b}, x_{1,1} \rangle^2 \geq \delta^2.$$

This implies that

$$[|\langle \beta, x_{a,b} \rangle| \leq 4\tau] \leq \frac{4e\tau}{\delta\sigma}.$$

Thus if we pick $\tau \leq \frac{\delta\sigma}{16ekl}$, taking a union bound we know that

$$\Pr\left[\forall a \in [k]\backslash[1], b \in [l] : |\langle \beta, x_{a,b} \rangle| \geq 4\tau\right] \geq \frac{1}{2}.$$

Moreover, since $\Pr[|\alpha| \leq \tau] \geq \frac{\tau}{e\sigma}$ and $\alpha$ is independent of $\beta$, we know that $\Pr[\mathcal{E}_\tau] \geq \frac{\tau}{16e^2\sigma}$.

The following proof will conditional on this event $\mathcal{E}_\tau$, and then treat $\beta$ as fixed and let $\alpha$ be the only random variable. In this way, we will have: for every $\alpha$ such that $|\alpha| \leq \tau$ and for every $a \in [k]\backslash[1], b \in [l]$, since $|\langle \beta, x_{a,b} \rangle| \geq 4\tau$ and $|\alpha\langle x_{1,1}, x_{a,b} \rangle| \leq \tau$,

$$\mathbf{ReLU}\left(\left\langle w_r^{(0)}, x_{a,b} \right\rangle\right) = (\alpha\langle x_{1,1}, x_{a,b} \rangle + \langle \beta, x_{a,b} \rangle) \mathbf{1}_{\langle \beta, x_{a,b} \rangle \geq 0}$$

which is a linear function of $\alpha$. With this information, we can rewrite $h\left(w_r^{(0)}\right)$ as:

$$h\left(w_r^{(0)}\right) = h(\alpha) := p_{1,1}v_{1,1,1}\mathbf{ReLU}(\alpha)$$
$$+ \sum_{b \in [l]\backslash[1]} p_{1,b}v_{1,1,b}\mathbf{ReLU}\left(\alpha\langle x_{1,1}, x_{1,b} \rangle + \langle \beta, x_{a,b} \rangle\right)$$
$$+ \mathsf{Linear}(\alpha)$$

where $p_{1,b}v_{1,1,b} \geq 0$ and $\mathsf{Linear}(\alpha)$ is some linear function in $\alpha$. Thus, we know that

$$\phi(\alpha) := p_{1,1}v_{1,1,1}\mathbf{ReLU}(\alpha) + \sum_{b \in [l]\backslash[1]} p_{1,b}v_{1,1,b}\mathbf{ReLU}\left(\alpha\langle x_{1,1}, x_{1,b} \rangle\right)$$

is a convex function with $|\partial_{\max}\phi(0) - \partial_{\min}\phi(0)| \geq v$. Then applying Lemma A.5 gives

$$\Pr_{\alpha \sim U(-\tau, \tau)}\left[|\phi(\alpha) + \mathsf{Linear}(\alpha)| \geq \frac{v\tau}{128}\right] \geq \frac{1}{16}.$$

Since for $\tau \leq \frac{\delta\sigma}{16ekl}$, conditional on $\mathcal{E}_\tau$ the density $p(\alpha) \in \left[\frac{1}{e\tau}, \frac{e}{\tau}\right]$, which implies that

$$\Pr\left[h\left(w_r^{(0)}\right) \geq \frac{v\tau}{128} \mid \mathcal{E}_\tau\right] \geq \frac{1}{16e}.$$

Thus we have:

$$\Pr\left[h\left(w_r^{(0)}\right) \geq \frac{v\tau}{128}\right] \geq \Pr\left[h\left(w_r^{(0)}\right) \geq \frac{v\tau}{128} \mid \mathcal{E}_\tau\right] \Pr[\mathcal{E}_\tau] = \Omega\left(\frac{\tau}{\sigma}\right). \tag{2}$$

Now we can look at $P_{1,r}$. By the random initialization of $w_r^{(0)}$, and since by our assumption $v_{1,a,b}, x_{a,b}$ are not functions of $w_r^{(0)}$, a standard tail bound of Gaussian random variables shows that for every fixed $v_{1,a,b}$ and every $c > 10$:

$$\Pr\left[h\left(w_r^{(0)}\right) \geq 10c\sigma\|P_{1,r}\|_2\right] = \Pr\left[\left\langle P_{1,r}, w_r^{(0)} \right\rangle \geq 10c\sigma\|P_{1,r}\|_2\right]$$
$$\leq e^{-c^2}.$$

Taking $c = 100\sqrt{\log \frac{kl}{\delta\sigma}}$ and putting together with inequality (2) with $\tau = \Theta\left(\frac{\delta\sigma}{kl}\right)$ complete the proof. $\qquad\square$

Now, we can take an epsilon net and switch the order of the quantifiers in Lemma A.2 as shown in the following lemma.

**Lemma A.3** (Lemma 5.2 restated). *For* $m = \tilde{\Omega}\left(\frac{k^3 l^2}{\delta}\right)$, *for every* $\{p_{a,b} v_{i,a,b}\}_{i,a\in[k],b\in[l]} \in [-v,v]$ *(that depends on* $w_r^{(0)}, a_{i,r}$, *etc.) with* $\max\{p_{a,b} v_{i,a,b}\}_{i,a\in[k],b\in[l]} = v$, *there exists at least* $\Omega(\frac{\delta}{kl})$ *fraction of* $r \in [m]$ *such that*

$$\left\|\frac{\tilde{\partial} L(w)}{\partial w_r}\right\|_2 = \tilde{\Omega}\left(\frac{v\delta}{kl}\right).$$

This lemma implies that if the classification error is large, then many $w_r$ will have a large gradient.

*Proof of Lemma A.3.* We first consider fixed $\{p_{a,b} v_{i,a,b}\}_{i,a\in[k],b\in[l]} \in [-v,v]$. First of all, using the randomness of $a_{i,r}$ we know that with probability at least $1/e$,

$$\left\|\frac{\tilde{\partial} L(w)}{\partial w_r}\right\|_2 = \left\|\sum_{i=1}^{k} a_{i,r} P_{i,r}\right\|_2 \geq \|P_{1,r}\|_2.$$

Now, apply Lemma A.2 we know that

$$\Pr\left[\left\|\sum_{i=1}^{k} P_{i,r}\right\|_2 = \tilde{\Omega}\left(\frac{v\delta}{kl}\right)\right] = \Omega\left(\frac{\delta}{kl}\right)$$

which implies that for fixed $\{p_{a,b} v_{i,a,b}\}_{i,a\in[k],b\in[l]} \in [-v,v]$ the probability that there are less than $O(\frac{\delta}{kl})$ of $r$ such that $\left\|\sum_{i=1}^{k} P_{i,r}\right\|_2$ is $\tilde{\Omega}\left(\frac{v\delta}{kl}\right)$ is no more than a value $p_{fix}$ given by:

$$p_{fix} \leq \exp\left\{-\Omega\left(\frac{\delta m}{kl}\right)\right\}.$$

Moreover, for every $\varepsilon > 0$, for two different $\{p_{a,b} v_{i,a,b}\}_{i,a\in[k],b\in[l]}, \{p_{a,b} v'_{i,a,b}\}_{i,a\in[k],b\in[l]} \in [-v,v]$ such that for all $i \in [k], a \in [k], b \in [l]$: $|p_{a,b} v_{i,a,b} - p_{a,b} v_{i,a,b}| \leq \varepsilon$. Moreover, since w.h.p. we know that every $|a_{i,r}| \leq L = \tilde{O}(1)$, it shows:

$$\left\|\sum_{a\in[k],b\in[l],i\in[k]} p_{a,b} a_{i,r}(v_{i,a,b} - v'_{i,a,b}) \mathbb{1}_{\left\langle w_r^{(0)}, x_{a,b}\right\rangle \geq 0} x_{a,b}\right\|_2 \leq L\varepsilon = \tilde{O}(\varepsilon)$$

which implies that we can take an $\ell_\infty$ $\varepsilon$-net over $\{p_{a,b} v_{i,a,b}\}_{i,a\in[k],b\in[l]} \in [-v,v]$ with $\varepsilon = \tilde{\Theta}\left(\frac{v\delta}{kl}\right)$. Thus, the probability that there exists $\{p_{a,b} v_{i,a,b}\}_{i,a\in[k],b\in[l]} \in [-v,v]$, such that there are no more than $O(\frac{\delta}{kl})$ fraction of $r \in [m]$ with $\left\|\sum_{i=1}^{k} P_{i,r}\right\|_2 = \tilde{\Omega}\left(\frac{v\delta}{kl}\right)$ is no more than:

$$p \leq p_{fix}\left(\frac{v}{\varepsilon}\right)^{k^2 l} \leq \exp\left\{-\Omega\left(\frac{\delta m}{kl}\right) + k^2 l \log \frac{v}{\varepsilon}\right\}.$$

With $m = \tilde{\Omega}\left(\frac{k^3 l^2}{\delta}\right)$ we complete the proof. $\qquad\square$

## A.3 Convergence

Having the lemmas, we can now prove the convergence:

**Lemma A.4** (Convergence). *Let us denote* $\max\{p_{a,b}v^{(t)}_{i,a,b}\} = v^{(t)}$. *Then for a sufficiently small* $\eta$, *we have that for every* $T = \tilde{\Theta}\left(\frac{\sigma\delta}{kl\eta}\right)$,

$$\frac{1}{T}\sum_{t=1}^{T}\left(v^{(t)}\right)^2 = \tilde{O}\left(\frac{k^5 l^5}{\delta^4 \sigma m}\right).$$

By our choice of $\sigma = \tilde{O}\left(\frac{1}{m^{1/2}}\right)$, we know that

$$\frac{1}{T}\sum_{t=1}^{T}\left(v^{(t)}\right)^2 = \tilde{O}\left(\frac{k^5 l^5}{\delta^4 m^{1/2}}\right)$$

Thus, this lemma shows that eventually $v^{(t)}$ will be small. However, we do not give any bound on how small the step size $\eta$ needs to be, and how a small $v^{(t)}$ leads to a small classification error. These are addressed in the proof of the general case in the next section, but here we are content with an eventually small $v^{(t)}$ for a sufficiently small $\eta$.

*Proof of Lemma A.4.* By Lemma A.3, we know that there are at least $\Omega\left(\frac{\delta}{kl}\right)$ fraction of $r \in [m]$ such that

$$\left\|\frac{\tilde{\partial}L(w^{(t)})}{\partial w_r}\right\|_2 = \tilde{\Omega}\left(\frac{v^{(t)}\delta}{kl}\right).$$

Now combine with Lemma A.1. If we pick $\tau = O\left(\frac{\sigma\delta}{k^2 l^2}\right)$, then at least $\Omega\left(\frac{\delta}{kl}\right)$ fraction of $r \in [m]$ have

$$\left\|\frac{\partial L(w^{(t)})}{\partial w_r}\right\|_2 = \tilde{\Omega}\left(\frac{v^{(t)}\delta}{kl}\right).$$

Thus, for a sufficiently small $\eta$, we have:

$$L(w^{(t)}) - L(w^{(t+1)}) = \eta\tilde{\Omega}\left(\left(\frac{v^{(t)}\delta}{kl}\right)^2\frac{\delta m}{kl}\right).$$

By the property of the initialization, we know that $L(w^{(0)}) = \tilde{O}(1)$. This implies that for every $t = \tilde{O}\left(\frac{\tau}{\eta}\right) = \tilde{O}\left(\frac{\sigma\delta}{k^2 l^2 \eta}\right)$ we have:

$$\sum_{s=1}^{t}\left(v^{(s)}\right)^2 = \tilde{O}\left(\frac{k^3 l^3}{\delta^3 \eta m}\right).$$

Now, we can take $T = \tilde{\Theta}\left(\frac{\sigma\delta}{k^2 l^2 \eta}\right)$ to obtain

$$\frac{1}{T}\sum_{t=1}^{T}\left(v^{(t)}\right)^2 = \tilde{O}\left(\frac{k^5 l^5}{\delta^4 \sigma m}\right).$$

This completes the proof. □

## A.4 Technical Lemmas

The following lemma above non-smooth convex function v.s. linear function is needed in the proof.

**Lemma A.5.** *Let* $\phi : \mathbb{R} \to \mathbb{R}$ *be a convex function that is non-smooth at* $0$. *Let* $\partial\phi(0)$ *be the set of partial gradient of* $\phi$ *at* $0$. *Define*

$$\partial_{\max}\phi(0) = \max\{\partial\phi(0)\}, \quad \partial_{\min}\phi(0) = \min\{\partial\phi(0)\}.$$

*We have for every $\tau \geq 0$, for every linear function $l(\alpha)$:*

$$\int_{-\tau}^{\tau} |\phi(\alpha) - l(\alpha)| d\alpha \geq \frac{\tau^2(\partial_{\max}\phi(0) - \partial_{\min}\phi(0))}{8}.$$

*Moreover,*

$$\Pr_{\alpha \sim U(-\tau,\tau)} \left[ |\phi(\alpha) - l(\alpha)| \geq \frac{\tau(\partial_{\max}\phi(0) - \partial_{\min}\phi(0))}{128} \right] \geq \frac{1}{16}.$$

*Proof of Lemma A.5.* Without loss of generality (up to subtracting a linear function on $\phi$), let us assume that $\phi(0) = 0$ and $l(\alpha) = -b$.

Moreover, denote $\rho = \partial_{\max}\phi(0) - \partial_{\min}\phi(0) \geq 0$, we know that at least one of the following is true:

1. $\partial_{\max}\phi(0) \geq \frac{\rho}{2}$,

2. $\partial_{\min}\phi(0) \leq -\frac{\rho}{2}$.

We shall give the proof for the case $\partial_{\max}\phi(0) \geq \frac{\rho}{2}$. The other case follows from replacing $\phi$ with $-\phi$.

Let us then consider the following two cases.

1. $b > 0$, in this case, by convexity of $\phi(\alpha)$ we have that $\forall \alpha > 0 : \phi(\alpha) > 0$. Thus,

$$\int_{-\tau}^{\tau} |\phi(\alpha) - l(\alpha)| d\alpha \geq \int_0^{\tau} \phi(\alpha) d\alpha \geq \frac{\rho}{4}\tau^2$$

2. $b < 0$, in this case, $\phi(\alpha)$ intersects with 0 at a point $\alpha_0 \geq 0$. Consider two cases:

   (a) $\alpha_0 \geq \frac{\tau}{2}$, then we have: $b \leq -\frac{\rho\tau}{4}$. Thus,

   $$\int_{-\tau}^{\tau} |\phi(\alpha) - l(\alpha)| d\alpha \geq \int_0^{\min\{\alpha_0,\tau\}} -\phi(\alpha) d\alpha \geq \frac{\rho}{8}\tau^2$$

   (b) $\alpha_0 \leq \frac{\tau}{2}$, then we have:

   $$\int_{-\tau}^{\tau} |\phi(\alpha) - l(\alpha)| d\alpha \geq \int_{\alpha_0}^{\tau} \phi(\alpha) d\alpha \geq \frac{\rho}{8}\tau^2$$

This completes the proof of the first claim. For the second claim, in case 1, we know that every $\alpha \in [\tau/2, \tau]$ would have $|\phi(\alpha) - l(\alpha)| \geq \frac{\tau\rho}{128}$. In case 2(a), every $\alpha \in [0, \alpha_0 - \tau/4]$ satisfies this claim. In case 2(b) we can take every $\alpha \in [\alpha_0 + \tau/4, \tau]$. This completes the proof. $\square$

## B  Proofs for the General Case

Recall that the loss is

$$L(w) = \frac{1}{N} \sum_{s=1}^N L(w, x_s, y_s)$$

where

$$L(w, x_s, y_s) = -\log o_{y_s}(x_s, w), \quad \text{where}$$

$$o_y(x, w) = \frac{e^{f_y(x,w)}}{\sum_{i=1}^k e^{f_i(x,w)}}.$$

We consider a minibatch SGD of batch size $B$, number of iterations $T = N/B$ and learning rate $\eta$ as the following process: Randomly divide the total training examples into $T$ batches, each of size $B$. Let the indices of the examples in the $t$-th batch be $\mathcal{B}_t$. The update rule is:

$$w_r^{(t+1)} = w_r^{(t)} - \eta \frac{1}{B} \sum_{s \in \mathcal{B}_t} \frac{\partial L(w^{(t)}, x_s, y_s)}{\partial w_r^{(t)}}, \forall r \in [m], \text{ where}$$

$$\frac{\partial L(w, x_s, y_s)}{\partial w_r} = \left( \sum_{i \neq y_s} a_{i,r} o_i(x_s, w) - \sum_{i \neq y_s} a_{y_s,r} o_i(x_s, w) \right) 1_{\langle w_r, x_s \rangle \geq 0} x_s.$$

The pseudo gradient on a point $(x_s, y_s)$ is defined as:

$$\frac{\tilde{\partial} L(w, x_s, y_s)}{\partial w_r} = \left( \sum_{i \neq y_s} a_{i,r} o_i(x_s, w) - \sum_{i \neq y_s} a_{y_s,r} o_i(x_s, w) \right) 1_{\langle w_r^{(0)}, x_s \rangle \geq 0} x_s.$$

The expected pseudo gradient is:

$$\frac{\tilde{\partial} L(w)}{\partial w_r} = \mathbb{E}_{(x_s, y_s)} \left[ \frac{\tilde{\partial} L(w, x_s, y_s)}{\partial w_r} \right].$$

In the following subsections, we first show that the gradient is coupled with the pseudo gradient, then show that if the classification error is large then the pseudo gradient is large, and finally prove the convergence.

## B.1 Coupling

We have the following lemma for coupling, analog to Lemma A.1.

**Lemma B.1** (Coupling). *For every unit vector $x \in \mathbb{R}^d$, w.h.p. over the random initialization, for every $\tau > 0$, for every $t = \tilde{O}\left(\frac{\tau}{\eta}\right)$ we have that for at least $1 - \frac{10\tau}{\sigma}$ fraction of $r \in [m]$:*

$$\frac{\partial L(w^{(t)}, x, y)}{\partial w_r} = \frac{\tilde{\partial} L(w^{(t)}, x, y)}{\partial w_r} (\forall y \in [k]), \quad \text{and} \quad |\langle w_r^{(t)}, x \rangle| \geq \tau.$$

*Proof.* The proof follows that for Lemma A.1. $\qquad \square$

## B.2 Expected Error Large $\implies$ Gradient Large

Following the same structure as before, we can write the expected pseudo gradient as:

$$\frac{\tilde{\partial} L(w)}{\partial \pi_r} = \sum_{i \in [k]} a_{i,r} P_{i,r}$$

where

$$P_{i,r} = \sum_{a \in [k], b \in [l]} p_{a,b} \mathbb{E}_{x_{a,b} \sim \mathcal{D}_{a,b}} \left[ v_{i,a,b}(x_{a,b}, w) 1_{\langle w_r^{(0)}, x_{a,b} \rangle \geq 0} x_{a,b} \right]$$

where $v_{s,a,b}(x_{a,b}, w)$ is defined as:

$$v_{s,a,b}(x_{a,b}, w) = \begin{cases} \frac{\sum_{i \neq a} e^{f_i(x_{a,b}, w)}}{\sum_{i=1}^{k} e^{f_i(x_{a,b}, w)}} & \text{if } s = a; \\ -\frac{e^{f_s(x_{a,b}, w)}}{\sum_{i=1}^{k} e^{f_i(x_{a,b}, w)}} & \text{otherwise.} \end{cases}$$

When clear from the context, we use $v_{s,a,b}(x_{a,b}, w)$ for short. When the choice of $x_{a,b}$ is not important, we will also use $v_{s,a,b}$.

We would like to show that if some $\mathbb{E}[p_{a,b}v_{i,a,b}]$ is large, a good fraction of $r \in [m]$ will have large pseudo gradient. Now, the first step is to show that for any fixed $\{p_{a,b}v_{i,a,b}\}$ (that does not depend on the random initialization $w_r^{(0)}$), with good probability (over the random choice of $w_r^{(0)}$) we have that $P_{i,r}$ is large; see Lemma B.2. Then we will take a union bound over an epsilon net on $\{p_{a,b}v_{i,a,b}\}$ to show that for every $\{p_{a,b}v_{,ia,b}\}$ (that can depend on $w_r^{(0)}$), at least a good fraction of of $P_{i,r}$ is large; See Lemma B.3.

**Lemma B.2** (The geometry of **ReLU**). *For any possible fixed set $\{p_{a,b}v_{1,a,b}\}$ (that does not depend on $w_r^{(0)}$) such that $\mathbb{E}[p_{1,1}v_{1,1,1}] = \max\{\mathbb{E}[p_{a,b}v_{1,a,b}]\}_{a\in[k],b\in[\ell]} = v$, we have:*

$$\Pr\left[\|P_{1,r}\|_2 = \tilde{\Omega}\left(\frac{v\delta}{kl}\right)\right] = \Omega\left(\frac{\delta}{kl}\right).$$

*Proof of Lemma B.2.* The proof is very similar to the proof of Lemma A.2.

We will actually prove that

$$h\left(w_r^{(0)}\right) = \sum_{a\in[k],b\in[l]} \mathbb{E}\left[p_{a,b}v_{1,a,b}\mathbf{ReLU}\left(\left\langle w_r^{(0)}, x_{a,b}\right\rangle\right)\right]$$

is large with good probability.

Let us denote $x_{a,b}^* = \frac{\mathbb{E}_{x_{a,b}\sim\mathcal{D}_{a,b}}[x_{a,b}]}{\|\mathbb{E}_{x_{a,b}\sim\mathcal{D}_{a,b}}[x_{a,b}]\|_2}$. Thus, we can decompose $w_r^{(0)}$ into:

$$w_r^{(0)} = \alpha x_{1,1}^* + \beta$$

where $\beta \perp x_{1,1}^*$. For every $\tau \geq 0$, consider the event $\mathcal{E}_\tau$ defined as

    1. $|\alpha| \leq \tau$.

    2.

$$\sum_{a\in[k]\setminus[1],b\in[l]} |p_{a,b}v_{1,a,b}|1_{|\langle\beta,x_{a,b}^*\rangle|\leq 4\tau} \leq \frac{v}{3}.$$

By the definition of initialization $w_r^{(0)}$, we know that:

$$\alpha \sim \mathcal{N}(0,\sigma^2)$$

and

$$\langle\beta, x_{a,b}^*\rangle \sim \mathcal{N}(0, (1 - \langle x_{a,b}^*, x_{1,1}^*\rangle^2)\sigma^2).$$

By assumption we can simply calculate that for every $a \in [k]\setminus[1], b \in [l]$: $1 - \langle x_{a,b}^*, x_{1,1}^*\rangle^2 \geq \delta^2$. This implies that

$$\mathbb{E}\left[1_{|\langle\beta,x_{a,b}^*\rangle|\leq 4\tau}\right] \leq \frac{4\tau}{\delta\sigma}.$$

Thus,

$$\sum_{a\in[k]\setminus[1],b\in[l]} \mathbb{E}\left[|p_{a,b}v_{1,a,b}|1_{|\langle\beta,x_{a,b}^*\rangle|\leq 4\tau}\right] \leq \frac{4\tau}{\delta\sigma}vl.$$

With $\tau = \frac{\sigma\delta}{12l}$, we know that $\Pr[\mathcal{E}_\tau] = \Omega\left(\frac{\tau}{\sigma}\right)$. The following proof will conditional on this event $\mathcal{E}_\tau$, and then treat $\beta$ as fixed and let $\alpha$ be the only random variable. In this way, for every $\alpha$ such that $|\alpha| \leq \tau$ and for every $a \in [k]\setminus[1], b \in [l]$:

$$\left\langle w_r^{(0)}, x_{a,b}\right\rangle = \alpha\langle x_{1,1}^*, x_{a,b}\rangle + \langle\beta, x_{a,b}\rangle$$

$$= \alpha\langle x_{1,1}^*, x_{a,b}^*\rangle + \langle\beta, x_{a,b}^*\rangle + \langle w_r^{(0)}, x_{a,b} - x_{a,b}^*\rangle.$$

With $|\alpha\langle x^*_{1,1}, x^*_{a,b}\rangle| \leq \tau$, and since $\mathbb{E}[\langle w_r^{(0)}, x_{a,b} - x^*_{a,b}\rangle] \leq \frac{3}{2}\sigma\lambda\delta < 2\tau$, we know that if $\left|\left\langle \beta, x^*_{a,b}\right\rangle\right| \geq 4\tau$, then

$$\mathbf{ReLU}\left(\left\langle w_r^{(0)}, x_{a,b}\right\rangle\right) = \left(\alpha\langle x^*_{1,1}, x_{a,b}\rangle + \langle\beta, x_{a,b}\rangle\right)1_{\langle\beta, x^*_{a,b}\rangle\geq 0}$$

is a linear function for $\alpha \in [-\tau, \tau]$ with probability $\geq 2/3$.

With this information, we can rewrite $h\left(w_r^{(0)}\right)$ as:

$$h\left(w_r^{(0)}\right) = h(\alpha) := \mathbb{E}\left[p_{1,1}v_{1,1,1}\mathbf{ReLU}\left(\alpha\langle x_{1,1}, x^*_{1,1}\rangle + \langle\beta, x^*_{1,1} - x_{1,1}\rangle\right)\right]$$
$$+ \sum_{b\geq 2}\mathbb{E}\left[p_{1,b}v_{1,1,b}\mathbf{ReLU}\left(\langle\alpha x^*_{1,1} + \beta, x_{a,b}\rangle\right)\right] + l(\alpha).$$

where $l(\alpha)$ is a convex function with $\partial_{\max}l(\tau) - \partial_{\max}l(-\tau) \leq v/3$.

This time, we know that w.h.p. $\langle\beta, x^*_{1,1} - x_{1,1}\rangle = \tilde{O}(\sigma\lambda\delta) \leq \tau/4$. This implies that for function $\phi$ defined as

$$\phi(\alpha) := \mathbb{E}\left[p_{1,1}v_{1,1,1}\mathbf{ReLU}\left(\alpha\langle x_{1,1}, x^*_{1,1}\rangle + \langle\beta, x^*_{1,1} - x_{1,1}\rangle\right)\right]$$
$$+ \sum_{b\geq 2}\mathbb{E}\left[p_{1,b}v_{1,1,b}\mathbf{ReLU}\left(\langle\alpha x^*_{1,1} + \beta, x_{a,b}\rangle\right)\right],$$

We will have $\partial_{\max}\phi(\tau/2) - \partial_{\max}\phi(-\tau/2) \geq v/2$. Now apply Lemma B.5, we can conclude from the same proof of Lemma A.2. $\square$

Now we can take the union bound to switch the order of quantifiers. However, we cannot do a naive union bound since there are infinitely many $x_{a,b}$. Instead, we will use a sampling trick to prove the following Lemma:

**Lemma B.3.** *For every $v > 0$, for $m = \tilde{\Omega}\left(\left(\frac{kl}{v\delta}\right)^4\right)$, for every possible $\{p_{a,b}v_{i,a,b}\}$ (that depend on $a_{i,r}, w_r^{(0)}$, etc.) such that $\max\{\mathbb{E}[p_{a,b}v_{i,a,b}]\}_{i,a\in[k],b\in[l]} = v$, there exists at least $\Omega\left(\frac{\delta}{kl}\right)$ fraction of $r \in [m]$ such that*

$$\left\|\frac{\hat{\partial}L(w)}{\partial w_r}\right\|_2 = \tilde{\Omega}\left(\frac{v\delta}{kl}\right).$$

This lemma implies that if the classification error is large, then many $w_r$'s have a large pseudo gradient.

*Proof of Lemma A.3.* We first pick $S$ samples $\mathcal{S} = \{x^{(s)}_{a,b}\}$, with $p_{a,b}S$ many from distribution $\mathcal{D}_{a,b}$, and with the corresponding value function $v^{(s)}_{i,a,b}$. Since each $v^{(s)}_{i,a,b} \in [-1, 1]$, we know that w.h.p., for every $i \in [k], a \in [k], b \in [l]$:

$$\left|\mathbb{E}[p_{a,b}v_{i,a,b}] - \frac{1}{p_{a,b}S}\sum_s p_{a,b}v^{(s)}_{i,a,b}\right| = \tilde{O}\left(\frac{1}{\sqrt{p_{a,b}S}}\right).$$

This implies that as long as $S = \tilde{\Omega}\left(\frac{1}{v^2}\right)$, we will have that

$$\max_{i\in[k],a\in[k],b\in[l]}\left\{\frac{1}{p_{a,b}S}\sum_s p_{a,b}v^{(s)}_{i,a,b}\right\} \in \left[\frac{1}{2}v, \frac{3}{2}v\right].$$

Thus, following the same proof as in Lemma A.3, but this time applying a union bound over $v^{(s)}_{i,a,b}$, we know that as long as $m = \tilde{\Omega}\left(\frac{Sk^2l}{\delta}\right)$, w.h.p. for every possible choices of $v^{(s)}_{i,a,b}$, there are at least

$\Omega\left(\frac{\delta}{kl}\right)$ fraction of $r \in [m]$ such that

$$\left\|\frac{1}{S}\sum_{x_{a,b}\in\mathcal{S}}\frac{\tilde{\partial}L(w,x_{a,b},a)}{\partial w_r}\right\|_2 = \tilde{\Omega}\left(\frac{v\delta}{kl}\right).$$

Now we consider the difference between the sample gradient and the expected gradient. Since $\left\|\frac{\tilde{\partial}L(w,x,y)}{\partial w_r}\right\|_2 \leq \tilde{O}(1)$, by standard concentration bound we know that w.h.p. for every $r \in [m]$,

$$\left\|\frac{1}{S}\sum_{x_{a,b}\in\mathcal{S}}\frac{\tilde{\partial}L(w,x_{a,b},a)}{\partial w_r} - \frac{\tilde{\partial}L(w)}{\partial w_r}\right\|_2 = \tilde{O}\left(\frac{1}{\sqrt{S}}\right).$$

This implies that as long as $S = \tilde{\Omega}\left(\left(\frac{kl}{v\delta}\right)^2\right)$, such $r \in [m]$ also have:

$$\left\|\frac{\tilde{\partial}L(w)}{\partial w_r}\right\|_2 = \tilde{\Omega}\left(\frac{v\delta}{kl}\right)$$

which completes the proof. □

## B.3 Convergence

We now show the following important lemma about convergence.

**Lemma B.4** (Convergence). *Denote* $\max\{\mathbb{E}[p_{a,b}v_{i,a,b}(x_{a,b},w^{(t)})]\}_{i,a\in[k],b\in[\ell]} = v^{(t)} = v$, *and let* $\gamma = \Omega\left(\frac{\delta}{kl}\right)$. *Then for a sufficiently small* $\eta = \tilde{O}\left(\frac{\gamma}{m}\left(\frac{v\delta}{kl}\right)^2\right)$, *if we run SGD with a batch size at least* $B_t = \tilde{\Omega}\left(\left(\frac{kl}{v\delta}\right)^4\frac{1}{\gamma^2}\right)$ *and* $t = \tilde{O}\left(\left(\frac{v\delta}{kl}\right)^2\frac{\sigma\gamma}{\eta}\right)$, *then w.h.p.,*

$$L(w^{(t)}) - L(w^{(t+1)}) = \eta\gamma m\tilde{\Omega}\left(\left(\frac{v\delta}{kl}\right)^2\right).$$

*Proof of Lemma B.4.* We know that for at least $\gamma$ fraction of $r \in [m]$ such that

$$\left\|\frac{\tilde{\partial}L(w^{(t)})}{\partial w_r}\right\|_2 = \tilde{\Omega}\left(\frac{v\delta}{kl}\right).$$

Note that w.h.p. over the random initialization, for every $(x,y)$, $\left\|\frac{\tilde{\partial}L(w^{(t)},x,y)}{\partial w_r}\right\|_2 \leq \tilde{O}(1)$. By Hoeffding concentration, this implies that for a randomly sampled batch $\mathcal{B}_t = \{(x_1,y_1),\cdots,(x_{B_t},y_{B_t})\}$ of size $B_t$, we have that w.h.p. over $\mathcal{B}_t$,

$$\left\|\frac{1}{B_t}\sum_{i=1}^{B_t}\frac{\tilde{\partial}L(w^{(t)},x_i,y_i)}{\partial w_r}\right\| = \tilde{\Omega}\left(\frac{v\delta}{kl}\right) - O\left(\frac{L}{\sqrt{B_t}}\right) = \tilde{\Omega}\left(\frac{v\delta}{kl}\right).$$

On the other hand, according to Lemma B.1 with $\tau = \frac{\sigma\gamma}{100B_t}$, we know that w.h.p. over the random initialization, for *every* $x_i$ in $\mathcal{B}_t$, we have: for at least $1-\gamma/(2B_t)$ fraction of $r \in [m]$, $\frac{\partial L(w^{(t)},x_i,y_i)}{\partial w_r} = \frac{\tilde{\partial}L(w^{(t)},x_i,y_i)}{\partial w_r}$. This implies that for at least $\gamma/2$ fraction of $r \in [m]$ such that for *every* $x_i$ in $\mathcal{B}_t$ we have $\frac{\partial L(w^{(t)},x_i)}{\partial w_r} = \frac{\tilde{\partial}L(w^{(t)},x_i)}{\partial w_r}$. Let us denote the set of these $r$ as set $\mathcal{R}$. Then for every $r \in \mathcal{R}$:

$$\left\|\frac{1}{B_t}\sum_{i=1}^{B_t}\frac{\partial L(w^{(t)},x_i,y_i)}{\partial w_r}\right\| = \tilde{\Omega}\left(\frac{v\delta}{kl}\right).$$

For every $r \in [m]$, let us denote $\tilde{\nabla}_{t,r} = \frac{1}{B_t} \sum_{i=1}^{B_t} \frac{\partial L(w^{(t)}, x_i, y_i)}{\partial w_r}$, and $\nabla_{t,r} = \frac{\partial L(w^{(t)})}{\partial w_r}$. Then similarly as above, since $\left\| \frac{\partial L(w^{(t)}, x, y)}{\partial w_r} \right\|_2 \leq \tilde{O}(1)$, by Hoeffding concentration, we have

$$\|\nabla_{t,r} - \tilde{\nabla}_{t,r}\|_2 = \tilde{O}\left(\frac{1}{\sqrt{B_t}}\right),$$

$$\|\nabla_{t,r}\|_2 = \tilde{\Omega}\left(\frac{v\delta}{kl}\right) - \tilde{O}\left(\frac{1}{\sqrt{B_t}}\right) = \tilde{\Omega}\left(\frac{v\delta}{kl}\right).$$

Now we consider the non-smooth gradient descent. Consider a newly sampled point $(x', y')$, and let us denote

$$\tilde{\nabla}'_{t,r} = \frac{\partial L(w^{(t)}, x', y')}{\partial w_r}.$$

By Lemma B.1, we know that w.h.p. over the random initialization, at least $1 - \frac{10\tau}{\sigma}$ fraction of $r$ satisfies $\langle w_r, x' \rangle \geq \tau$. Let us denote the set of these $r$'s as $\mathcal{S}_r$. We know that on these sets, the function is $\tilde{O}(1)$ smooth and $\tilde{O}(1)$ Lipschitz smooth. By Lemma B.6,

$$\Delta_t := L(w^{(t)} - \eta\tilde{\nabla}_t, x', y') - L(w^{(t)}, x', y')$$

$$\leq -\eta \sum_{r \in \mathcal{S}_r} \langle \tilde{\nabla}_{t,r}, \tilde{\nabla}'_{t,r} \rangle + \sum_{r \in [m] \setminus \mathcal{S}_r} \tilde{O}(\eta) + \tilde{O}(\eta^2 m^2)$$

$$\leq -\eta \sum_{r \in [m]} \langle \tilde{\nabla}_{t,r}, \tilde{\nabla}'_{t,r} \rangle + \tilde{O}\left(\frac{\eta\tau m}{\sigma}\right) + \tilde{O}(\eta^2 m^2). \tag{3}$$

Let $G_1$ denote the event that (3) holds.

Note that w.h.p. over the random initialization, $|L(w^{(t)}, x', y')| = \tilde{O}(L\eta tmk) = \tilde{O}(m)$, and $\|\tilde{\nabla}_{t,r,i}\| \leq \tilde{O}(1)$, $\|\tilde{\nabla}'_{t,r}\| \leq \tilde{O}(1)$ for all $(x_i, y_i)$'s and $(x', y')$. Let $G_0$ denote this event.

Then we have $P[\neg G_0]$ and $P[\neg G_1]$ bounded by $1/\text{poly}(k, l, m, 1/\delta, 1/\epsilon)$. Conditioned on $G_0$, we have $\nabla_{t,r} = \mathbb{E}_{(x', y')}\left[\tilde{\nabla}'_{t,r}|G_0\right]$ and $L(w^{(t)}) - L(w^{(t+1)}) = \mathbb{E}_{(x', y')}[\Delta_t|G_0]$ where the expectation is over $(x', y')$. Now we have

$$\nabla_{t,r} = \mathbb{E}_{(x', y')}\left[\tilde{\nabla}'_{t,r}|G_0, G_1\right] P[G_1|G_0] + \mathbb{E}_{(x', y')}\left[\tilde{\nabla}'_{t,r}|G_0, \neg G_1\right] P[\neg G_1|G_0].$$

So

$$\left\| \nabla_{t,r} - \mathbb{E}_{(x', y')}\left[\tilde{\nabla}'_{t,r}|G_0, G_1\right] \right\|_2 = \frac{1}{\text{poly}(k, l, m, 1/\delta, 1/\epsilon)}.$$

Then

$$L(w^{(t)}) - L(w^{(t+1)}) = \mathbb{E}_{(x', y')}[\Delta_t|G_0, G_1] P[G_1|G_0] + \mathbb{E}_{(x', y')}[\Delta_t|G_0, \neg G_1] P[\neg G_1|G_0]$$

$$\geq \frac{\eta}{2} \sum_{r \in [m]} \langle \tilde{\nabla}_{t,r}, \mathbb{E}_{(x', y')}\left[\tilde{\nabla}'_{t,r}|G_0, G_1\right] \rangle - \tilde{O}(\eta^2 m^2) - \tilde{O}\left(\frac{\eta\tau m}{\sigma}\right)$$

$$- \frac{\tilde{O}(m)}{\text{poly}(k, l, m, 1/\delta, 1/\epsilon)}$$

$$\geq \frac{\eta}{2} \sum_{r \in [m]} \langle \tilde{\nabla}_{t,r}, \nabla_{t,r} \rangle - \tilde{O}(\eta^2 m^2) - \tilde{O}\left(\frac{\eta\tau m}{\sigma}\right)$$

$$- \frac{\tilde{O}(m)}{\text{poly}(k, l, m, 1/\delta, 1/\epsilon)} - \frac{\tilde{O}(\eta m)}{\text{poly}(k, l, m, 1/\delta, 1/\epsilon)}$$

$$\geq \frac{\eta}{2} \sum_{r \in [m]} \langle \tilde{\nabla}_{t,r}, \nabla_{t,r} \rangle - \tilde{O}(\eta^2 m^2) - \tilde{O}\left(\frac{\eta\tau m}{\sigma}\right).$$

Note that $\tilde{\nabla}_{t,r}$ concentrates around $\nabla_{t,r}$. This leads to w.h.p. when $\eta = \tilde{O}\left(\frac{\gamma}{m}\left(\frac{v\delta}{kl}\right)^2\right)$, $\tau = \tilde{O}\left(\gamma\left(\frac{v\delta}{kl}\right)^2\sigma\right)$, and $B_t = \tilde{\Omega}\left(\left(\frac{kl}{v\delta}\right)^4\frac{1}{\gamma^2}\right)$,

$$L(w^{(t)}) - L(w^{(t+1)}) \geq \sum_{r=1}^{m} \frac{\eta}{2}\|\tilde{\nabla}_{t,r}\|_2^2 - \tilde{O}(\eta^2 m^2) - \tilde{O}\left(\frac{\eta\tau m}{\sigma}\right) - \eta\tilde{O}\left(\frac{m}{\sqrt{B_t}}\right)$$

$$\geq \eta\gamma m\tilde{\Omega}\left(\left(\frac{v\delta}{kl}\right)^2\right) - \tilde{O}(\eta^2 m^2) - \tilde{O}\left(\frac{\eta\tau m}{\sigma}\right) - \eta\tilde{O}\left(\frac{m}{\sqrt{B_t}}\right)$$

$$\geq \eta\gamma m\tilde{\Omega}\left(\left(\frac{v\delta}{kl}\right)^2\right).$$

This completes the proof. $\qquad\square$

Now we can prove the main theorem.

**Theorem 4.1.** *Suppose the assumptions (A1)(A2)(A3) are satisfied. Then for every $\varepsilon > 0$, there is $M = poly(k, l, 1/\delta, 1/\varepsilon)$ such that for every $m \geq M$, after doing a minibatch SGD with batch size $B = poly(k, l, 1/\delta, 1/\varepsilon, \log m)$ and learning rate $\eta = \frac{1}{m \cdot poly(k,l,1/\delta,1/\varepsilon,\log m)}$ for $T = poly(k, l, 1/\delta, 1/\varepsilon, \log m)$ iterations, with high probability:*

$$\Pr_{(x,y)\sim\mathcal{D}}\left[\forall j \in [k], j \neq y, f_y(x, w^{(T)}) > f_j(x, w^{(T)})\right] \geq 1 - \varepsilon.$$

*Proof of Theorem 4.1.* Let $v_{i,a,b}^{(t)}$ denote $v_{i,a,b}(x_{a,b}, w^{(t)})$.

First, we will show that if $\Pr_{(x,y)\sim\mathcal{D}}\left[\forall j \in [k], j \neq y, f_y(x, w^{(t)}) > f_j(x, w^{(t)})\right] \leq 1 - \varepsilon$, there must be one $a, b$ such that $\mathbb{E}[v_{i,a,b}^{(t)}] \geq \varepsilon^2$. Let us denote $\max\{\mathbb{E}[p_{a,b}v_{i,a,b}^{(t)}]\} = v^{(t)} = v$. For a particular $a \in [k], b \in [l]$, for any $x_{a,b}$ from $\mathcal{D}_{a,b}$, by definition,

$$v_{a,a,b}(x_{a,b}, w^{(t)}) = 1 - \frac{e^{f_a(x_{a,b}, w^{(t)})}}{\sum_{i=1}^{k} e^{f_i(x_{a,b}, w^{(t)})}}.$$

Then for every $\varepsilon \leq \frac{1}{e}$, if $v_{a,a,b}^{(t)}(x_{a,b}, w^{(t)}) \leq \varepsilon$, then

$$\forall i \in [k], i \neq a : f_a(x_{a,b}, w^{(t)}) \geq f_i(x_{a,b}, w^{(t)}) + 1,$$

which implies that the prediction is correct. So if $\mathbb{E}[v_{a,a,b}^{(t)}] \leq \varepsilon^2$, then there are at most $\varepsilon$ fraction of $x_{a,b}$ such that $f_a(x_{a,b}, w^{(t)}) \leq f_i(x_{a,b}, w^{(t)})$ for some $i \neq a$. In other words, if $\Pr_{(x,y)\sim\mathcal{D}}\left[\forall j \in [k], j \neq y, f_y(x, w^{(t)}) > f_j(x, w^{(t)})\right] \leq 1 - \varepsilon$, there must be some $i, a, b$ such that $\mathbb{E}[v_{i,a,b}^{(t)}] \geq \varepsilon^2$.

Now, consider two cases:

1. $p_{a,b} \leq \frac{\varepsilon}{2kl}$. For all such $a, b$, even if all the predictions are wrong, it will only increase the total error by $\varepsilon/2$ so the other half $\varepsilon/2$ error must come from other $p_{a,b}$.

2. $p_{a,b} \geq \frac{\varepsilon}{2kl}$, which means that $\mathbb{E}[p_{a,b}v_{i,a,b}^{(t)}] \geq \frac{\varepsilon}{2kl}\mathbb{E}[v_{i,a,b}^{(t)}] \geq \frac{\varepsilon^3}{8kl}$. Thus, $\max\{\mathbb{E}[p_{a,b}v_{i,a,b}^{(t)}]\} = v^{(t)} = v \geq \frac{\varepsilon^3}{8kl}$.

Therefore, to prove the theorem, it suffices to show that $v^{(t)}$ will be smaller than $\frac{\varepsilon^3}{8kl}$ after a proper amount of iterations. Suppose $v^{(t)} \geq \frac{\varepsilon^3}{8kl}$, then by Lemma B.4, as long as

$$t = \tilde{O}\left(\frac{\sigma}{\eta}\frac{\delta^3\varepsilon^6}{k^5\ell^5}\right), \tag{4}$$

we have:
$$L(w^{(t)}) - L(w^{(t+1)}) \geq \tilde{O}\left(\eta m \frac{\delta^3 \varepsilon^6}{k^5 \ell^5}\right).$$

Note that by the random initialization, originally for each $f_i$ we have: for every unit vector $x \in \mathbb{R}^d$, $\langle w_r^{(0)}, x \rangle \sim \mathcal{N}(0, \sigma^2)$. Thus, with $\sigma = \frac{1}{\sqrt{m}}$ and $a_{i,r} \sim \mathcal{N}(0, 1)$, an elementary calculation shows that w.h.p.,

$$|f_i(x, w^{(0)})| = \left| \sum_{r \in [m]} a_{i,r} \mathbf{ReLU}(\langle w_r^{(0)}, x \rangle) \right| = \tilde{O}(1).$$

Thus, $L(w^{(0)}) = \tilde{O}(1)$. Since $L(w) \geq 0$, we know that $L(w^{(t)}) - L(w^{(t+1)}) \geq \tilde{O}\left(\eta m \frac{\delta^3 \varepsilon^6}{k^5 \ell^5}\right)$ can happen for at most

$$\tilde{O}\left(\frac{1}{\eta m} \frac{k^5 \ell^5}{\delta^3 \varepsilon^6}\right)$$

iterations. By our choice of $\eta$, we know that $\eta m = \tilde{O}\left(\frac{\delta^3 \varepsilon^6}{k^5 \ell^5}\right)$, so we need at most $T = \tilde{O}\left(\frac{k^{10} \ell^{10}}{\delta^6 \varepsilon^{12}}\right)$ iterations.

To this end, we just need

$$\frac{\sigma}{\eta} \frac{\delta^3 \varepsilon^6}{k^5 \ell^5} = \tilde{\Omega}\left(\frac{1}{\eta m} \frac{k^5 \ell^5}{\delta^3 \varepsilon^6}\right)$$

to make sure (4) holds so that we can keep the coupling before convergence. This is true as long as $m = \tilde{\Omega}\left(\frac{k^{20} \ell^{20}}{\delta^{12} \varepsilon^{24}}\right)$. □

## B.4 Technical Lemmas

The following lemma above non-smooth convex function v.s. linear function is needed in the proof.

**Lemma B.5.** *Let $\phi : \mathbb{R} \to \mathbb{R}$ be a convex function. Let $\partial \phi(x)$ be the set of partial gradient of $\phi$ at $x$. Define*

$$\partial_{\max} \phi(x) = \max\{\partial \phi(x)\}, \quad \partial_{\min} \phi(x) = \min\{\partial \phi(x)\}.$$

*We have that for every $\tau \geq 0$, for every convex function $l(\alpha)$, let $\gamma = (\partial_{\max} \phi(\tau/2) - \partial_{\min} \phi(-\tau/2)) - (\partial_{\max} l(\tau) - \partial_{\min} l(-\tau))$, then*

$$\int_{-\tau}^{\tau} |\phi(\alpha) - l(\alpha)| d\alpha \geq \frac{\tau^2 \gamma}{32}$$

*and*

$$\Pr_{a \sim U(-\tau, \tau)} \left[ |\phi(\alpha) - l(\alpha)| \geq \frac{\tau \gamma}{512} \right] \geq \frac{1}{64}.$$

*Proof.* Without loss of generality, we can assume that either $\partial_{\max} l(\tau)$ and $\partial_{\max} \phi(\tau/2) \geq \gamma/2$, or $\partial_{\min} l(-\tau) = 0$ and $\partial_{\min} \phi(-\tau/2) \leq -\gamma/2$. The lemma can be proved using the same argument as in Lemma A.5. □

We also need the following lemma regarding the gradient descent on non-smooth function.

**Lemma B.6.** *Suppose for every $i \in [m]$, $g_i : \mathbb{R}^d \to \mathbb{R}$ is a L-Lipschitz smooth function. Moreover, suppose for an $r \in [m]$, for all $i \in [m-r]$ we have that $g_i$ is also L-smooth. Suppose $g : \mathbb{R} \to \mathbb{R}$ is L-smooth and L-Lipschitz smooth, and let $f(w)$ denote $g(\sum_{i \in [m]} g_i(w_i))$. Then for every $w, \delta \in \mathbb{R}^{dm}$ with $\|\delta_i\|_2 \leq p$ we have:*

$$g\left(\sum_{i \in [m]} g_i(w_i + \delta_i)\right) - g\left(\sum_{i \in [m]} g_i(w_i)\right) \leq \sum_{i \in [m-r]} \left\langle \frac{\partial f(w)}{\partial w_i}, \delta_i \right\rangle + L^3 m^2 p^2 + L^2 rp.$$

*Proof of Lemma B.6.* The proof of this lemma follows directly from

$$g\left(\sum_{i\in[m]} g_i(w_i+\delta_i)\right) - g\left(\sum_{i\in[m]} g_i(w_i)\right)$$

$$\leq g\left(\sum_{i\in[m-r]} g_i(w_i+\delta_i) + \sum_{i>m-r} g_i(w_i)\right) - g\left(\sum_{i\in[m]} g_i(w_i)\right)$$

$$+ L\left|\sum_{i>m-r} g_i(w_i) - \sum_{i>m-r} g_i(w_i+\delta_i)\right|$$

$$\leq g\left(\sum_{i\in[m-r]} g_i(w_i+\delta_i) + \sum_{i>m-r} g_i(w_i)\right) - g\left(\sum_{i\in[m]} g_i(w_i)\right) + L^2 pr$$

$$\leq \left\langle \nabla g\left(\sum_{i\in[m]} g_i(w_i)\right), \sum_{i\in[m-r]} g_i(w_i+\delta_i) - \sum_{i\in[m-r]} g_i(w_i)\right\rangle$$

$$+ \frac{L}{2}\left\|\sum_{i\in[m-r]} g_i(w_i+\delta_i) - \sum_{i\in[m-r]} g_i(w_i)\right\|^2 + L^2 pr$$

$$\leq \left\langle \nabla g\left(\sum_{i\in[m]} g_i(w_i)\right), \sum_{i\in[m-r]} g_i(w_i+\delta_i) - \sum_{i\in[m-r]} g_i(w_i)\right\rangle + L^3 m^2 p^2 + L^2 pr$$

$$\leq \sum_{i\in[m-r]} \left\langle \frac{\partial f(w)}{\partial w_i}, \delta_i\right\rangle + L^3 m^2 p^2 + L^2 pr$$

where the last line follows from the chain rule and Lipschitz smoothness, and the last to second line follows from

$$\left|\sum_{i\in[m-r]} g_i(w_i+\delta_i) - \sum_{i\in[m-r]} g_i(w_i)\right| \leq Lpm.$$

This completes the proof. □

## C Illustration of the Separability Assumption

(a) Each class as two components      (b) Each class as one component

Figure 3: Illustration of the separability assumption. The data lie in $\mathcal{R}^2$ and are from two classes $-$ and $+$. The $+$ class contains points uniformly over two balls of diameter $1/10$ with centers $(0,0)$ and $(2,2)$, and the $-$ class contains points uniformly over two balls of the same diameter with centers $(0,2)$ and $(2,0)$. (a) We can view each ball in each class as one component, then the data will satisfy the separability assumption with $\ell = 2$. (b) We can also view each class as just one component, but the data will not satisfy the separability assumption with $\ell = 1$.

Recall the separability assumption introduced in Section 3:

**(A1)** (Separability) There exists $\delta > 0$ such that for every $i_1 \neq i_2 \in [k]$ and every $j_1, j_2 \in [l]$,
$$\mathrm{dist}\left(\mathrm{supp}(\mathcal{D}_{i_1,j_1}), \mathrm{supp}(\mathcal{D}_{i_2,j_2})\right) \geq \delta.$$
Moreover, for every $i \in [k], j \in [l]$,
$$\mathrm{diam}(\mathrm{supp}(\mathcal{D}_{i,j})) \leq \lambda\delta, \text{ for } \lambda \leq 1/(8l).$$

In this assumption, each class can contain multiple components when $\ell \geq 2$. This allows more flexibility and also allows non-linearly separable data. See Figure 3 for such an example. The data lie in $\mathcal{R}^2$ and are from two classes $-$ and $+$. The $+$ class contains points uniformly over two balls of diameter $1/10$ with centers $(0,0)$ and $(2,2)$, and the $-$ class contains points uniformly over two balls of the same diameter with centers $(0,2)$ and $(2,0)$. As illustrated in Figure 3(a), the data satisfy the separability assumption with $\ell = 2$: each ball in each class is viewed as one component, then the distance between any two points in one component is at most $1/10$ while the distance between any two points from different components will be at least $19/10$. However, as illustrated in Figure 3(b), the data do not satisfy the separability assumption with $\ell = 1$, by viewing each class as just one component. This demonstrates that allowing $\ell \geq 2$ leads to more flexibility. Furthermore, the data are clearly not linearly separable, showing that the assumption captures nonlinear structures of practical data better than linear separability.

## D Additional Experimental Results

Here we provide some additional experimental results.

### D.1 Statistics When Achieving A Small Error v.s. Number of Hidden Nodes

Recall that our analysis that for a learning rate decreasing with the number of hidden nodes $m$, the number of iterations to get the accuracy roughly remain the same. A more direct way to check is to plot the number of steps to achieve the accuracy for different $m$. As shown in Figure 4, the number of steps roughly match what our theory predicts.

Furthermore, Figure 5 shows the relative distances when achieving the desired accuracies. It is observed that the distances scale roughly as $O(1/\sqrt{m})$. In particular, they closely match $2/3\sqrt{m}$ on the synthetic data and $1/6\sqrt{m}$ on MNIST (the red lines in the figures), where $m$ is the number of hidden nodes. Explanations are left for future work.

| (a) On synthetic data | (b) on MNIST |

Figure 4: Number of steps to achieve $98\%$ on the synthetic data and $95\%$ test accuracy on MNIST for different values of number of hidden nodes. They are roughly the same for different number of hidden nodes.

| (a) On synthetic data | (b) on MNIST |

Figure 5: Relative distances when achieving $98\%$ on the synthetic data and $95\%$ test accuracy on MNIST for different values of number of hidden nodes. They closely match $2/3\sqrt{m}$ on the synthetic data and $1/6\sqrt{m}$ on MNIST (the red lines), where $m$ is the number of hidden nodes.

## D.2 Synthetic Data with Larger Variances

Here we test the effect of the in-component variance on the learning process. First recall that the synthetic data are of 1000 dimension and consist of $k = 10$ classes, each having $\ell = 2$ components. Each component is of equal probability $1/(kl)$, and is a Gaussian with covariance $\sigma/\sqrt{d}I$ and its mean is i.i.d. sampled from a Gaussian distribution $\mathcal{N}(0, \sigma_0/\sqrt{d})$. 1000 training data points and 1000 test data points are sampled. Here we fix $\sigma_0 = 5$ and vary $\sigma$ and plot the test accuracy, the coupling, the distance across different time steps, and the spectrum of the final solution.

Figure 6 shows that the test accuracy decreases with increasing variance $\sigma$, and it takes longer time to get a good solution. On the other hand, an increasing variance does not change the trends for activation patterns, distance, and the rank of the weight matrix. This is possibly due to that the signal in the updates remain small with increasing variances, while the noise in the updates act similarly as the randomness in the weights.

(a) Test accuracy

(b) Coupling

(c) Distance from the initialization

(d) Rank of accumulated updates ($y$-axis in log-scale)

Figure 6: Results for synthetic data with different variances.

## D.3 Synthetic Data with Larger Number of Components in Each Class

Here we test the effect of the number of components in each class on the learning process. First recall that the synthetic data are of 1000 dimension and consist of $k = 10$ classes, each having $\ell$ components. Each component is of equal probability $1/(kl)$, and is a Gaussian with covariance $1/\sqrt{d}I$ and its mean is i.i.d. sampled from a Gaussian distribution $\mathcal{N}(0, 5/\sqrt{d})$. 1000 training data points and 1000 test data points are sampled. Here we vary $\ell$ from 1 to 7 and plot the test accuracy, the coupling, the distance across different time steps, and the spectrum of the final solution.

Figure 7 shows that the test accuracy decreases with increasing number of components $\ell$ in each class, and it takes longer time to get a good solution. On the other hand, a larger $\ell$ leads to more significant coupling and smaller relative distances at the same time step. This is probably because the learning makes less progress due to the more complicated structure of the data.

(a) Test accuracy

(b) Coupling

(c) Distance from the initialization

(d) Rank of accumulated updates for 4 components in each class ($y$-axis in log-scale)

Figure 7: Results for synthetic data with larger number of components in each class.