[Reviews · NeurIPS 2018]

Reviewer 1



This paper studies learning over-parametrized single hidden layer ReLU neural networks for multi-class classification via SGD and the corresponding generalization error. They consider a mixture data distribution where each class has well-separated and compact support. The authors show SGD applied on the considered learning model achieves good prediction error with high probability under suitable assumptions. As a result even in severely over-parametrized models, SGD can generalize well although the network has enough capacity to fit arbitrary labels. The main insight in the theoretical analysis appears to be the observation that in the over-parametrized case, many ReLU neurons don't change their activation pattern when initialized randomly. Although the claimed result is interesting, there are some issues in clarity, presentation and notation which needs to be addressed. I have some doubts on Lemma 5.1 (please see below). The numerical evidence presented is weak and not clearly explained. Please find below a detailed list of comments and suggestions. I hope that the authors can clarify these points in the revision. 1. page 5 line 170. It's not clear why p_{a,b} appears in the expression for L(w). Is this a population level quantity ? Please refrain from overloading notation (L(w)). 2. In lemma 5.1 it appears that \tau needs to be small to have a large fraction of indices to match the pseudo ReLu network. Is tau = O(1) or tau*k*l = O(1)? 3. If you combine all l distributions in each class to a single distribution, isn't the separability assumption still valid ? But the sample complexities depend on l, which becomes smaller when merging distributions. Can you please clarify ? 4. In the numerical experiments, it is claimed that over-parametrization helps in optimization. However this is not immediately evident from the figures. In Figure 2 the test accuracy vs iteration plots appear to be overlapping and Figure 1 is hard to read. 5. page 3 line 101. Can you add a reference citing which previous works consider l=1 in this setting ? 6. page 3 line 115. Can you clearly define L(x_i,w) ? If possible, please use different notation for L(x_i,w) since L(w) is defined and refrain from overloading. 7. page 5 line 181. Typo 'randomness'

Reviewer 2



This paper studies the optimization of one-hidden-layer neural network for solving multi-class classification problems with the softmax loss. It shows that with a large enough over-parametrization (that depends polynomially on dimension factors and 1/\eps), mini-batch SGD converges to a classifier with at least 1-\eps classification accuracy. I feel like this is a strong paper. The part which surprised me the most is Lemma 5.2, which says that with enough over-parametrization, whenever the "pseudo" classification error is still large, the gradient norm of the pseudo-net is also large, so that SGD will keep making progress. This is a joint statement on approximation theory (there has to be a point with low classification error) and on the optimization landscape (if you're not there your gradient has to be large). This joint statement seems too strong to be true generically but here it probably comes from the linearity of the pseudo-net. Combining with Lemma 5.1, which couples the original-net and the pseudo-net for a bounded number of iterations (T <= O(1/\eta)), one gets that SGD on the original-net has to make progress unless the classification error is low, which is the main result. This entire analysis pattern is pretty novel and I have not seen similar analyses in the literature. The discussion session is very well-written and insightful as well. In particular, pretty inspiringly, it postulates that with over-parametrized models, there can be a solution not far from the initialization with high probability. This phenomenon is also presented in the experiments. I wonder that whether the relative Frobenious error scales as 1/sqrt{K} -- it looks like this is the case for any fixed t. As a potential exploration, it might be good to try to choose different t for different K based on some pre-determined metric (e.g. classification accuracy > 90%) and see the scale of the relative error, to see if this phenomenon is more convincing. One limitation is that the setting might make the classification problem itself easy (each class is mixture of L well-separated components). But given the novelty of the result and the analysis, I think it brings in interesting new ideas and directions for future work, which makes it a nice contribution.

Reviewer 3



This paper analyzes the generalization performance of two-layer ReLU networks in an overparametrized regime. Specifically, the networks are trained with SGD over separable multiclass problems. In such setting, the authors prove that the learned network has small generalization error with high probability. This result generalizes the result in [7], which only considers linearly separable data. This paper is well organized and easy to follow. The proof sketch and discussion on insights make it easy to understand the technique in use and the implications of this work. There is no doubt that the authors address an important problem. Explaining the generalization performance of deep learning is a very core task of the modern machine learning. But the contribution of this work seems incremental. The theoretical result of this paper generalizes the result in [7] by adopting a more general separable dataset. It is true that such structured (multiclass) data is closer to the practical scenario. While the truly practical scenarios normally involve non-separable dataset, which cannot be easily analyzed with the approach used in the paper (the separability parameter \delta > 0 seems necessary even for the simplified case in Section 5). Moreover, mixing the generalization and optimization in the main theorem likely complicates the analysis and restricts the scope of the result. In particular, SGD might have nothing special in regard to learning a solution with good generalization performance. It just happens to be the only practical method for neural network training. For random forest [a] and kernel machine [b], solutions that are not learned by SGD (e.g. exact solution for kernel regression) can achieve both (almost) zero training error and excellent generalization performance. Note that all these models/architectures are overparametrized. So we see empirically that for several overparametrized architectures, methods other than SGD can learn a solution with a small generalization error. Thus, it is hard to tell which inductive bias is more important to obtain the good generalization performance, the one derived from the optimization method or the one enforced by the architecture selection. Some minor things: Line 123: or every \epsilon ---> for every \epsilon. Line 141: by SGD ---> by GD. Line 276: converges to 0 ---> converges to 100%. [a] Wyner, Abraham J., et al. "Explaining the success of adaboost and random forests as interpolating classifiers." [b] Belkin, Mikhail, et al. "To understand deep learning we need to understand kernel learning." ===== After Rebuttal ===== After reading the rebuttal and all reviews, I feel that this paper deserves a higher score.

Reviewer 4



Summary: This paper analyzes SGD on cross-entropy objective for a 2 layer neural network with ReLU activations. They assume a separability condition on the input data. Pros: 1. This paper analyzes a very interesting version of the problem I believe. That is, they stick to the cross-entropy loss and they study SGD the algorithm used most often in practice. 2. The techniques seems novel and innovative which is different from past work. I think this is an important contribution and adds strength to this work. Questions: 1. The assumption A1 seems very similar to linear separability. I don't understand why one cannot club D_{i,j} for all j \in l into a single distribution D_i. Then assumption A1 is the same as separability. Am I missing something? It would be helpful to explain the assumption A1 pictorially. 2. The rates in Theorem 4.1 should be made explicit. Currently it is extremely opaque and hard to interpret the result and verify its correctness. 3. The idea of a pseudo loss and coupling that to the true loss is interesting. I think it would improve the presentation of the paper to add a bit more intuition of the technique of page 5.